# Towards All-Atom Foundation Models for Biomolecular Binding Affinity Prediction

**Liang Shi**[1,*]  **Zuobai Zhang**[2,3,*]  **Huiyu Cai**[2,3]

**Santiago Miret**[4]  **Zhi Yang**[1,†]  **Jian Tang**[2,5,6,†]

[1]School of Computer Science, Peking University  [2]Mila - Québec AI Institute
[3]Université de Montréal  [4]Intel Labs  [5]HEC Montréal  [6]CIFAR AI Chair

liang.shi@stu.pku.edu.cn
zuobai.zhang@mila.quebec
yangzhi@pku.edu.cn
jian.tang@hec.ca

## Abstract

Biomolecular interactions play a critical role in biological processes. While recent breakthroughs like AlphaFold 3 have enabled accurate modeling of biomolecular complex structures, predicting binding affinity remains challenging mainly due to limited high-quality data. Recent methods are often specialized for specific types of biomolecular interactions, limiting their generalizability. In this work, we adapt AlphaFold 3 for representation learning to predict binding affinity, a non-trivial task that requires shifting from generative structure prediction to encoding observed geometry, simplifying the heavily conditioned trunk module, and designing a framework to jointly capture sequence and structural information. To address these challenges, we introduce the **A**tom-level **Di**ffusion **T**ransformer (**ADiT**), which takes sequence and structure as inputs, employs a unified tokenization scheme, integrates diffusion transformers, and removes dependencies on multiple sequence alignments and templates. We pre-train three ADiT variants on the PDB dataset with a denoising objective and evaluate them across protein-ligand, drug-target, protein-protein, and antibody-antigen interactions. The model achieves state-of-the-art or competitive performance across benchmarks, scales effectively with model size, and successfully identifies wet-lab validated affinity-enhancing antibody mutations, establishing a generalizable framework for biomolecular interactions. Our open-source implementation is available at https://github.com/VectorShi/ADiT.

## 1 Introduction

Biomolecular interactions, including protein-protein and protein-ligand interactions, are pivotal to a wide range of biological processes, such as cellular functions (Busch et al., 2025) and therapeutic applications (Pacholarz et al., 2012; Cheng et al., 2021). Recent breakthroughs, exemplified by AlphaFold 3 and related methods (Abramson et al., 2024; Krishna et al., 2024; Chai Discovery, 2024; Chen et al., 2025; Wohlwend et al., 2024), have enabled accurate structure modeling of complex biological proteins. For example, AlphaFold 3 demonstrates remarkable accuracy in predicting protein complex structures from sequences, effectively resolving both the spatial arrangement and interaction mechanisms of protein-protein and protein-ligand interactions. Despite recent research progress, structure prediction remains an intermediate step, with the ultimate goal being the design of functional proteins with strong binding affinities for specific targets.

Binding affinity prediction remains difficult due to the scarcity of reliable experimental affinity labels. Early methods relied exclusively on sequences as input (Sun et al., 2017; Rao et al., 2019b), failing to fully leverage recent advancements in structure prediction (Abramson et al., 2024), even though

---

*Equal contribution.
†Corresponding authors.

all-atom structures are fundamental to understanding binding mechanisms and determining affinity. Recent approaches have incorporated structural information but are often specialized for specific types of biomolecular interactions. For instance, RDE-Network (Luo et al., 2023), DiffAffinity (Liu et al., 2024), and Prompt-DDG (Wu et al., 2024) target protein-protein binding affinity, while MGraph-DTA (Yang et al., 2022), HGNN-DTA (He et al., 2023), and ProFSA (Gao et al., 2023) focus on protein-ligand binding affinity. This narrow specialization limits the generalizability of these approaches and their applicability across diverse interaction types and data modalities.

Recent advancements in foundation models across various domains, such as BERT (Kenton & Toutanova, 2019) and the GPT series (Floridi & Chiriatti, 2020; Achiam et al., 2023) for language, the SAM series (Kirillov et al., 2023; Ravi et al., 2024) for vision, and CLIP (Radford et al., 2021) for multi-modality, have inspired new directions for addressing these challenges. These models are first pre-trained on broad datasets using self-supervised learning objectives and then adapted to a wide range of downstream tasks (Bommasani et al., 2021). By reducing the need for domain-specific inductive biases and leveraging large datasets, these models learn powerful representations that significantly enhance generalization capabilities across diverse tasks. In biology, AlphaFold 3 exemplifies this paradigm by developing a universal model capable of predicting various types of biomolecular complex structures, demonstrating the power of generalization across interaction types.

Inspired by these successes, we adapt the AlphaFold 3 architecture for representation learning to design a structure-based unified foundation model for binding affinity prediction. While AlphaFold 3 is primarily a generative model for predicting biomolecular structures, adopting AlphaFold 3 for affinity prediction requires non-trivial modifications. We propose a novel strategy to adapt AlphaFold 3 as a representation learner, guided by three key insights: (i) when the model's goal shifts from *predicting geometry* to *encoding observed geometry*, the heavy conditioning (trunk) module becomes less critical and might allow for a more lightweight design; (ii) the transformer-based, atom and sequence-level architecture offers a general framework for jointly encoding both structural and sequential information; (iii) by large-scale pre-training on structure data, it can potentially mitigate data scarcity in affinity prediction and improve generalization across diverse interaction tasks.

Building on this framework, we present the **Atom-level Diffusion Transformer** (**ADiT**), a universal, all-atom structure foundation model designed to learn transferable representations across diverse biomolecular interactions. Our model accepts sequence and structure as inputs, employs a unified scheme to tokenize proteins and molecules, and generates atom-level, token-level, and complex-level representations. We integrate key components of diffusion transformers while simplifying the architecture by removing dependencies on multiple sequence alignments and templates. Following a pre-training and fine-tuning framework, we train three versions of our ADiT, namely ADiT-S, ADiT-M, and ADiT-L, on the PDB dataset. These pre-trained models are evaluated across four different types of interactions including protein-ligand, drug-target, protein-protein, and antibody-antigen, and demonstrate state-of-the-art or competitive performance across all benchmarks. Ablation studies highlight the importance of large-scale pre-training and atom-level modeling, while consistent performance gains with increasing model size align with established scaling trends in other domains.

**Main contributions:** *(i)* We present a general-purpose foundation model, ADiT, designed for binding affinity prediction across various types of biomolecular interactions. *(ii)* Extensive experiments demonstrate that ADiT, empowered by large-scale pre-training, achieves state-of-the-art or competitive performance across diverse, well-established benchmarks. *(iii)* We observe consistent performance improvements across various benchmarks as model size increases, reflecting scaling trends observed in other domains. *(iv)* A case study on antibodies shows that ADiT can successfully identify affinity-enhancing mutations, with predictions validated by wet-lab experiment results.

## 2 RELATED WORK

**Protein Representation Learning.** Research in protein representation learning can be broadly categorized into two main paradigms: sequence-based methods and structure-based methods. Sequence-based approaches interpret protein sequences as a form of biological language, leveraging large-scale pre-training on sequence or alignment datasets to extract functional and evolutionary information (Rao et al., 2019b;a; Elnaggar et al., 2021; Rives et al., 2021; Meier et al., 2021; Wang et al., 2024b; Xu et al., 2023). In contrast, structure-based representation learning has gained significant attention due to the critical role of protein structures in determining function, as well as recent breakthroughs in structure prediction tools (Jumper et al., 2021). These methods explicitly

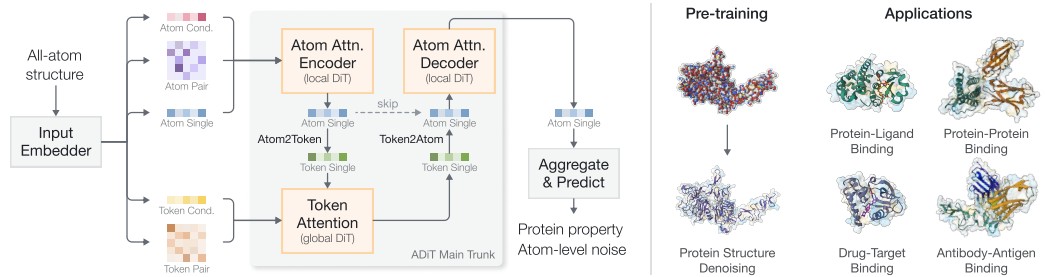

Figure 1: **Biomolecular complex representation learning with ADiT.** The model inputs all-atom structures, processes them through a multi-level attention trunk, and generates atom representations. It is pre-trained using a denoising objective and then fine-tuned for downstream application tasks.

encode protein structures using techniques such as 3D CNNs (Derevyanko et al., 2018), GNNs (Zhang et al., 2023c; Zhao et al., 2024; Fan et al., 2023), and transformers (Jiao et al., 2024). They employ various self-supervised learning strategies, including contrastive learning, self-prediction and diffusion denoising, to train structure encoders effectively (Zhang et al., 2023c; Chen et al., 2022; Zhang et al., 2023d). Additionally, some approaches use structure tokenizers to convert 3D protein structures into token sequences, enabling the application of sequence-based methods for structural representation learning (Su et al., 2024; Wang et al., 2024a; Heinzinger et al., 2024). However, many current approaches have two key limitations: (1) they primarily focus on coarse-grained, residue-level structures, and (2) they are restricted to proteins alone. In this work, we address these gaps by focusing on all-atom representation learning for general biomolecular complexes.

**Biomolecular Complex Modeling.** Recent advancements in molecular docking, both in predicting bound structures from unbound inputs (Ganea et al., 2022; Zhang et al., 2023a; Corso et al., 2023; Ketata et al., 2023) and in directly predicting bound structures from sequences (Abramson et al., 2024; Krishna et al., 2024; Chai Discovery, 2024; Chen et al., 2025; Wohlwend et al., 2024), have substantially advanced the field of biomolecular complex structure prediction. Despite these progress, predicting structures remains an intermediate step towards designing proteins with functional properties like strong binding affinity. Numerous studies have focused on predicting these binding affinities for protein-protein and protein-molecule interactions. For instance, prior work has explored energy-based methods (Delgado et al., 2019; Alford et al., 2017), evolution-based approaches (Meier et al., 2021; Rao et al., 2021; Notin et al., 2022), end-to-end learning models (Shan et al., 2022), and structure-informed pre-training methods (Yang et al., 2020; Luo et al., 2023; Wu et al., 2024; Mo et al., 2024; Liu et al., 2024) for protein-protein binding affinity. For protein-ligand binding, pre-training methods have become the dominant approach due to data scarcity (Karimi et al., 2019; Liu et al., 2023; Wu et al., 2022; Zhou et al., 2023; Gao et al., 2023). Protein-ligand affinity prediction is crucial in drug-target prediction (Yang et al., 2022; He et al., 2023; Kroll et al., 2023a;b), underscoring its importance in protein design. It is also common to benchmark against zero-shot baselines such as physics-based scoring functions (e.g., Rosetta (Barlow et al., 2018)), predicted mutational effects (e.g., NERE (Jin et al., 2023b) and DSMBind (Jin et al., 2023a)), and structural confidence metrics (e.g., AlphaFold 3's ipTM (Abramson et al., 2024)), although these typically fall far short of specialized fine-tuned models. Besides, Boltz-2 has been recently proposed for protein-ligand binding affinity prediction (Passaro et al., 2025). However, a unified framework for biomolecular binding affinity prediction remains elusive, with many studies still relying on sequence-based methods and overlooking breakthroughs like AlphaFold 3. In this work, we introduce a general foundation model for biomolecular complexes, leveraging structures to learn robust representations.

## 3 METHOD

In this section, we introduce the **Atom-level Diffusion Transformer** (**ADiT**), a unified foundation model for diverse biomolecular interactions. Our model employs a unified scheme to tokenize and featurize proteins and molecules (Sec. 3.2) and incorporates diffusion transformers to capture token- and atom-level interactions (Sec. 3.3). Following standard practices in modern foundation model development (Kenton & Toutanova, 2019; Achiam et al., 2023), we train ADiT with a pre-training and fine-tuning framework (Sec. 3.4). An overview of our method is shown in Figure 1.

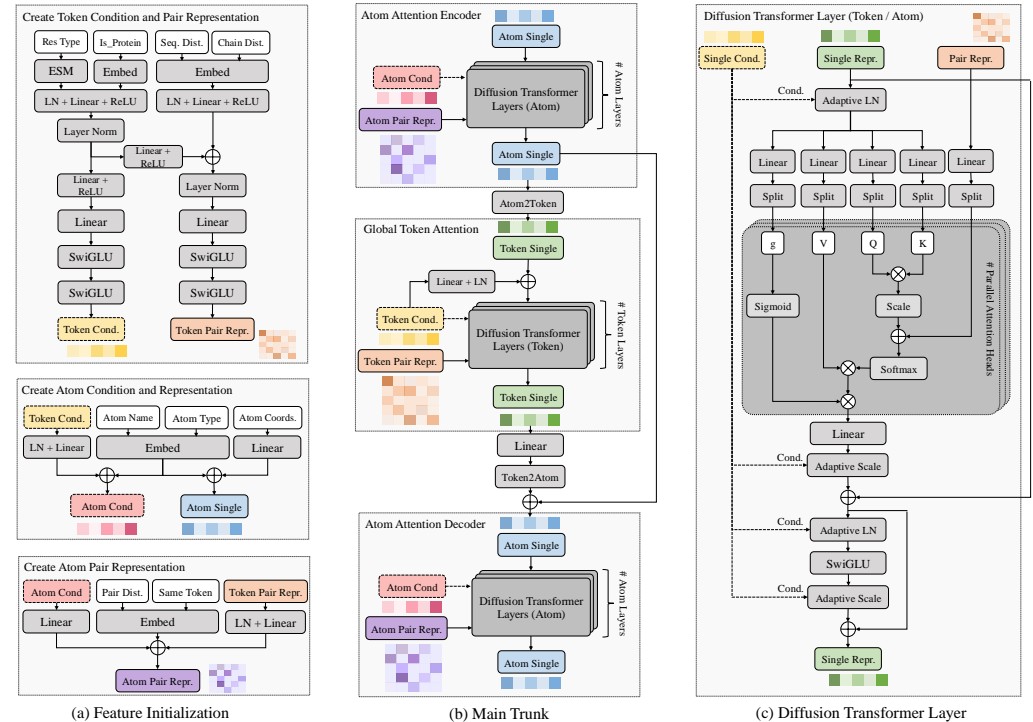

Figure 2: **The Atom-level Diffusion Transformer (ADiT) architecture**. *(a)* Initialization of features for atom and token sequences. *(b)* Main trunk for hierarchical representation learning, comprising an atom attention encoder, global token attention, and an atom attention decoder. *(c)* Diffusion transformer blocks with multiple sparse attention update layers for atom-level modeling.

## 3.1 ALPHAFOLD FOR BINDING AFFINITY PREDICTION

Given the sequence $A \in \{1, .., 20\}^L$ and the structure $\mathbf{x} \in \mathbb{R}^{L \times 3}$ of a biomolecular complex, our objective is to extract atom representations and use them to predict binding affinity through a downstream linear or muli-layer perceptron (MLP) prediction head. As AlphaFold 3 (Abramson et al., 2024) provides a highly expressive, transformer-based architecture that jointly encodes sequential and structural information at the atom level across diverse types of biomolecular interactions, it is a natural starting point for building general-purpose representation learner.

However, adapting AlphaFold 3 for representation learning is both non-trivial and essential. AlphaFold 3 was originally designed as a generative model for structure prediction, and prior work has shown that generative models often produce suboptimal representations when applied directly to downstream tasks such as protein function or affinity prediction (Hu et al., 2022). This limitation arises because generative objectives primarily optimize for reconstructing structures rather than capturing features that are informative for functional or interaction properties. Similar trends have been observed in other domains: for instance, generative models for images or text, while capable of producing high-quality outputs, do not automatically yield strong embeddings for tasks like classification, retrieval, or zero-shot generalization (Li et al., 2023; Achiam et al., 2023). Consequently, transforming AlphaFold 3 into ADiT, a model explicitly designed for representation learning, requires substantial modifications both its architecture and pre-training strategy.

Specifically, AlphaFold 3 is a conditional diffusion model that consists of two primary components: a trunk conditioning module and a diffusion module heavily conditioned on diffusion time step. The conditioning module is a multimodal encoder that integrates information from sequences, multiple sequence alignments (MSAs), and structural templates for structure prediction. However, in representation learning scenarios where complex structures are provided as input, the model's primary role shifts from inferring geometry to encoding the observed geometry. As a result, the reliance on heavy multimodal conditioning becomes less critical. Additionally, AlphaFold 3 depends heavily on

MSAs and templates, which are not always available and may be suboptimal for learning generalizable representations. In contrast, the diffusion module employs a modern transformer architecture that is well-suited for scalability. It incorporates advanced techniques such as SwiGLU (Shazeer, 2020), gating mechanisms from AlphaFold 2 (Jumper et al., 2021), and a two-level architecture that alternates between atom-level and token-level representations. These features make the diffusion module particularly effective for updating representations based on both the geometry and the sequence. Based on these insights, we remove the dependencies on MSAs and templates, and introduce ADiT for representation learning across diverse biomolecular interactions and affinity prediction.

## 3.2 SIMPLIFIED FEATURIZATION SCHEME

We employ a generalized tokenization scheme, where each residue in proteins is represented as a single token, and each heavy atom in small molecules is encoded as a single token. We initialize the multi-level input features using a top-down approach. This involves first initializing features at the token level, followed by deriving atom-level features by integrating both atom-specific information and the propagated token-level features. Details of our feature embedder are illustrated in Fig. 2(a).

**Token Level Encoding.** We first construct a token-level sequence embedder that encodes the chemical and biological information of complexes, producing representations for all tokens. These representations serve as the conditioning input for the subsequent diffusion transformer. Unlike AlphaFold 3, which relies on MSA and template inputs, introducing additional complexity and potential challenges when MSAs or templates are unavailable or of low quality, we use a pre-trained protein language model, ESM-2-650M (Lin et al., 2023), to incorporate evolutionary information. Specifically, the token conditioning representations consist of two components: the ESM sequence feature and the token type embedding. The ESM sequence feature is only computed for residue tokens, while for small molecule tokens, this feature is set to zero. The token type embedding encodes the origin of the token, distinguishing between protein and small molecule tokens.

Next, we construct pair representations by concatenating the corresponding token conditioning representations and integrate relative position encodings that include both relative sequence and chain distance. Different from AlphaFold 3, we avoid computationally intensive Pairformer blocks, but only use a concatenation and linear layer for embedding to save compute. We use $c^{\text{token}}$ and $z^{\text{token}}$ to denote the token conditioning and pair representations, respectively.

**Atom Level Encoding.** At the atomic level, we encode both chemical and evolutionary information, along with structural information for each atom. Note that the subsequent diffusion transformer will condition on the former (chemical and evolutionary information) while extracting representations from the latter (structural information). To achieve this, we distinguish between the single representation $s^{\text{atom}}$, which includes structural information, and the conditions $c^{\text{atom}}$, which exclude it.

To generate atom conditioning representations $c^{\text{atom}}$, we embed the atom type and name (for protein atoms) and augment them with token conditioning representations. For single atom representations $s^{\text{atom}}$, we encode the atom type, name, and coordinates. Then, for atom pair representations $z^{\text{atom}}$, we combine atom conditioning representations, embed their Euclidean distance using RBF kernels, and encode whether they belong to the same token. These atom pair representations are then augmented with the corresponding token pair representations to capture both local and global interactions.

## 3.3 HIERARCHICAL REPRESENTATION LEARNING

To capture both token- and atom-level information, we adopt a hierarchical representation learning framework, utilizing Diffusion Transformer (DiT) modules (Peebles & Xie, 2023). The framework begins by updating atom representations through $N_{\text{block}}^{\text{atom}}$ DiT blocks, followed by an "Atom2Token" average pooling operation to generate token representations. These token representations are then refined using $N_{\text{block}}^{\text{token}}$ DiT blocks. Subsequently, a linear layer and a "Token2Atom" unpooling operation produce updated atom representations, which are combined with the previous atom representations via a skip connection. The "Token2Atom" operation is a non-learned broadcast operation. The atom representations are further refined through another $N_{\text{block}}^{\text{atom}}$ DiT blocks. An overview is illustrated in Figure 2(b) and outlined in Algorithm 1.

**Diffusion Transformer Block.** We adapt the diffusion transformer to model both interatomic and inter-token interactions for representation learning. The transformer comprises multiple stacked

blocks, each containing an Adaptive LayerNorm function, a multi-head self-attention module, and a transition function, as detailed in Algorithm 2. We add skip connections to enhance training stability. The Adaptive LayerNorm function and transition function process both single and conditioning representations, while the multi-head self-attention module computes attention weights using single and pair representations. Formally, the multi-head self-attention module is defined as:

$$\boldsymbol{q}_i^h, \boldsymbol{k}_i^h, \boldsymbol{v}_i^h \leftarrow \text{Linear}_{q,k,v}(\boldsymbol{s}_i), \quad A_{ij}^h \leftarrow \text{softmax}_j \left( \frac{\boldsymbol{q}_i^{h\top} \boldsymbol{k}_j^h}{\sqrt{d}} + \text{Linear}_b^h(\boldsymbol{z}_{ij}) + \beta_{ij} \right),$$

$$\boldsymbol{g}_i^h \leftarrow \text{sigmoid}(\text{Linear}_g(\boldsymbol{s}_i)), \quad \boldsymbol{s}_i \leftarrow \text{Linear}\left( \text{concat}_h \left( \boldsymbol{g}_i^h \odot \sum_j A_{ij}^h \boldsymbol{v}_j^h \right) \right),$$

where $h$ denotes the attention head index, and $d$ is the dimension of the latent representation. The term $\beta_{ij}$ controls whether interactions between pair $(i, j)$ are modeled. For token-level interactions, all $\beta_{ij}$ values are zero. For atom-level interactions, each group of 32 atoms attends to a nearby group of 128 atoms (based on sequence proximity). The DiT architecture is illustrated in Figure 2(c).

Notably, ADiT utilizes only a single linear layer to embed all atom positions during the initialization of atom representations, without incorporating geometric biases such as locality or SE(3) invariance. This non-equivariant design contrasts with current trends that emphasize stronger domain-specific inductive biases, thereby facilitating the general usage on different data modalities. We approximate the required invariance through a combination of centroid-centering of the input coordinates and data augmentation via random rotations during pre-training. We hypothesize that this non-equivariant approach offers greater flexibility to capture crucial non-geometric features (e.g., electrostatic interactions and chemical semantics from RDKit and ESM-2 features) that govern binding thermodynamics, without being overly constrained by the strong geometric bias of equivariant layers. Furthermore, non-equivariant Transformers are architecturally simpler and highly scalable, enabling us to build the foundation model more efficiently than current equivariant frameworks.

### 3.4 PRE-TRAINING AND FINE-TUNING FRAMEWORK

Inspired by recent foundation models, we adopt a two-stage training approach: first pre-training our model with a denoising objective, followed by fine-tuning it for specific downstream tasks.

**Pre-training.** Denoising pre-training has emerged as a widely used self-supervised learning approach for learning representations of 3D structures (Zaidi et al., 2023; Zhang et al., 2023b). The process involves adding Gaussian noise to each atom's coordinate as $\varepsilon \sim \mathcal{N}(\boldsymbol{0}, \sigma^2 \boldsymbol{I})$, where $\sigma$ represents the noise scale. The noisy structure is then input into ADiT, and a noise prediction head is applied to the resulting atom representations. As ADiT is a non-equivariant model, we incorporate random rotations of structures as data augmentation. Following (Zaidi et al., 2023), we use a fixed noise scale, treating it as a tunable hyperparameter. Preliminary experiments indicate that $\sigma = 0.5\text{Å}$ yields the best performance. Although we also experimented with varying noise scales as in (Zhang et al., 2023d), no practical improvements were observed. Nonetheless, we hypothesize that a carefully curated training distribution of noise scales could enable the model to better capture both coarse- and fine-grained features. Due to computational constraints, we leave this exploration to our future work.

**Fine-tuning.** To adapt our pre-trained model for downstream tasks, we fine-tune it using a reduced learning rate. For predicting properties of the bound complex, we input the structure into the model to obtain output atom representations. These representations are then sequentially aggregated through "Atom2Token" average pooling and "Token2Complex" sum pooling, followed by a prediction head. In addition, our goal is to learn representations from clean samples, rather than to generate samples from noise as in standard diffusion models. We do not need to condition on the diffusion time step. Accordingly, during fine-tuning, the time step is always set to zero, corresponding to a clean sample.

**Difference with AlphaFold 3.** While ADiT leverages architectural components similar to those in AlphaFold 3, our model represents a significant re-engineering tailored for the task of affinity prediction, leading to substantial differences in architecture, input, and objective. The key distinctions are as follows: (1) **Goal:** AlphaFold 3 is fundamentally a generative model designed for structure prediction, while ADiT is a representation learner specialized for affinity prediction. (2) **Inputs:** We remove AlphaFold 3's reliance on computationally expensive MSA and template conditioning, which we replace with pre-trained protein features from ESM-2 and explicit RDKit features for small molecules. (3) **Architecture:** We simplify the backbone by removing the heavy trunk module and the

computationally intensive Pairformer blocks, focusing instead on a more scalable stack of Diffusion Transformer blocks. (4) **Objective:** We adapt the core diffusion transformer module to a simpler denoising objective, rather than the generative structure prediction task. We believe these adaptations are important for the strong performance of ADiT on binding affinity prediction.

# 4 EXPERIMENTS

The goal of this paper is to develop a *general-purpose* foundation model for predicting the binding affinity of various types of biomolecular complexes. In this section, we evaluate our model on four downstream tasks and compare its performance against recent strong baselines. The experimental results demonstrate that as a general-purpose foundation model, ADiT achieves state-of-the-art or competitive results across these tasks, even compared with specialized models.

## 4.1 PRE-TRAINING SETUP

For pre-training, we curate datasets from the Protein Data Bank (PDB) (Berman et al., 2000), resulting in 433,297 protein single chains, 481,382 protein-protein interaction examples, and 427,947 protein-ligand interaction examples. Here the interaction example refers to the interaction between two specific chains that form chain-chain interfaces within a single PDB entry. Two chains are considered to interact if the minimum heavy-atom distance between them is less than 5 angstroms. To accelerate pre-training, we leverage the clustering results provided by the RCSB PDB, yielding a total of 150,009 clusters. Importantly, we do not use any function labels during pre-training to avoid potential data leakage and ensure a fair evaluation on downstream tasks.

To explore the scaling effects, we pre-trained three model variants: ADiT-S (12M params), ADiT-M (35M params), and ADiT-L (253M params). The largest variant, ADiT-L, matches the number of layers and hidden dimensions used in AlphaFold 3. The differences between these variants lie in the configurations of their atom and token transformer layers. For details, please refer to Appendix D.1.

## 4.2 PROTEIN-LIGAND BINDING AFFINITY PREDICTION

**Setup.** For protein-ligand binding affinity, we adopt the setup from Atom3D (Townshend et al., 2021), where each sample consists of the bound structure of a protein-ligand complex, and our goal is to predict its binding affinity. The dataset includes two splits based on protein sequence similarity: one with a max. sequence identity 30% (LBA-30) and another with a max. sequence identity 60% (LBA-60). We use the Root Mean Square Error (RMSE), Pearson correlation coefficient, and Spearman's rank correlation coefficient as evaluation metrics. We include recent strong baselines, and additionally fine-tune Protenix (Chen et al., 2025), an open-source reproduction of AlphaFold3.

**Results.** The results in Table 1 demonstrate that all three ADiT variants achieve competitive performance compared to previous strong baselines. Among them, ADiT-L achieves state-of-the-art results across both data splits and all evaluation metrics. Specifically, on the LBA-30 benchmark, it outperforms the previous best method, GET, with a 1.9% increase in Pearson correlation and a 0.77% improvement in Spearman correlation. On LBA-60 benchmark, it surpasses prior best baselines ProNet and ProFSA, achieving a 7.2% reduction in RMSE, along with 4.2% and 4.3% increases in Pearson and Spearman correlations, respectively. Notably, even the smallest variant, ADiT-S, with only 12M parameters, outperforms most existing baselines. Furthermore, we observe that prior methods exhibit limited generalization capabilities, performing well on either LBA-30 or LBA-60, but not both, *e.g.*, GET and ProNet. In contrast, our approach demonstrates consistent and robust performance across both dataset splits. Moreover, our results underscore the superiority of ADiT over Protenix across all evaluation metrics, showing that simply fine-tuning generative models for representation learning and downstream tasks is insufficient to unlock their full potential.

## 4.3 DRUG-TARGET BINDING AFFINITY PREDICTION

**Setup.** For the drug-target binding affinity task, we employ the widely used Davis dataset (Davis et al., 2011), consisting of 30,056 data points that capture binding affinities between 72 small-molecule drugs and 442 target protein kinases. Following (He et al., 2023), we perform a random split of the

Table 1: **Protein-ligand binding affinity prediction results.** The best result is highlighted in **bold**, and the second-best result is underlined. Differences within 0.002 are considered negligible.

| Model | Sequence Identity 30% | | | Sequence Identity 60% | | |
|---|---|---|---|---|---|---|
| | RMSE↓ | Pearson↑ | Spear.↑ | RMSE↓ | Pearson↑ | Spear.↑ |
| B & B (Bepler & Berger, 2019) | 1.985 | 0.165 | 0.152 | 1.891 | 0.249 | 0.275 |
| DeepDTA (Öztürk et al., 2018) | 1.866 | 0.472 | 0.471 | 1.762 | 0.666 | 0.663 |
| DeepAffnity (Karimi et al., 2019) | 1.893 | 0.415 | 0.426 | – | – | – |
| MaSIF (Gainza et al., 2020) | 1.484 | 0.467 | 0.455 | 1.426 | 0.709 | 0.701 |
| Atom3D-GNN (Townshend et al., 2021) | 1.601 | 0.545 | 0.533 | 1.408 | 0.743 | 0.743 |
| IEConv (Hermosilla et al., 2021) | 1.554 | 0.414 | 0.428 | 1.473 | 0.667 | 0.675 |
| Holoprot (Somnath et al., 2021) | 1.464 | 0.509 | 0.500 | 1.365 | 0.749 | 0.742 |
| ProNet (Wang et al., 2023) | 1.463 | 0.551 | 0.551 | 1.343 | 0.765 | 0.761 |
| Uni-Mol (Zhou et al., 2023) | 1.520 | 0.558 | 0.540 | 1.619 | 0.645 | 0.653 |
| GeoSSL (Liu et al., 2023) | 1.451 | 0.577 | 0.572 | – | – | – |
| ProFSA (Gao et al., 2023) | 1.377 | 0.628 | 0.620 | 1.377 | 0.764 | 0.762 |
| GET (Kong et al., 2024) | 1.327 | 0.620 | 0.611 | 1.510 | 0.675 | 0.688 |
| GET-PS (Kong et al., 2024) | **1.309** | 0.633 | 0.642 | 1.509 | 0.676 | 0.680 |
| Protenix (Chen et al., 2025) | 1.354 | 0.569 | 0.593 | 1.781 | 0.707 | 0.737 |
| **ADiT-S** | 1.337 | 0.626 | 0.618 | 1.413 | 0.740 | 0.740 |
| **ADiT-M** | 1.353 | 0.622 | 0.630 | 1.335 | 0.764 | 0.752 |
| **ADiT-L** | **1.308** | **0.645** | **0.647** | **1.246** | **0.797** | **0.795** |

Table 2: **Results for drug-target and protein-protein binding affinity prediction.** The best result is in **bold**, and the second-best is underlined. Differences within 0.002 are considered negligible.

| Drug-Target | | | Protein-Protein | | | | |
|---|---|---|---|---|---|---|---|
| Model | MSE↓ | $r_m^2$↑ | Model | Pearson↑ | Spear.↑ | RMSE↓ | MAE↓ |
| DeepDTA | 0.261 | 0.630 | ESM-IF | 0.319 | 0.280 | 1.886 | 1.285 |
| MT-DTI | 0.245 | 0.665 | End-to-End | 0.637 | 0.488 | 1.619 | 1.176 |
| GraphDTA | 0.229 | 0.685 | DDGPred | 0.658 | 0.468 | **1.499** | 1.082 |
| rzMLP | 0.205 | 0.709 | MIF-Network | 0.652 | 0.513 | 1.593 | 1.146 |
| EnsembleDLM | 0.202 | – | RDE-Network | 0.644 | 0.558 | 1.579 | 1.112 |
| FusionDTA | 0.208 | 0.743 | DiffAffinity | 0.669 | 0.556 | 1.535 | 1.093 |
| MgraphDTA | 0.207 | 0.710 | Prompt-DDG | 0.677 | **0.591** | 1.520 | **1.077** |
| NHGNN-DTA | **0.196** | 0.744 | Surface-VQMAE | 0.648 | 0.561 | 1.588 | 1.127 |
| **ADiT-S** | 0.252 | 0.690 | **ADiT-S** | 0.660 | 0.524 | 1.597 | 1.132 |
| **ADiT-M** | 0.216 | 0.734 | **ADiT-M** | 0.683 | 0.539 | 1.559 | 1.098 |
| **ADiT-L** | **0.198** | **0.751** | **ADiT-L** | **0.691** | 0.560 | 1.540 | 1.088 |

Davis dataset five times and report the average performance across these splits. We employ two evaluation metrics: Mean Squared Error (MSE) and the $r_m^2$ metric. Details in Appendix D.3.

**Results.** The results, summarized in the left part of Table 2, demonstrate that ADiT delivers competitive performance, with its best-performing variant, ADiT-L, achieving state-of-the-art results on both the MSE and $r_m^2$ metrics. Moreover, ADiT exhibits a clear scaling effect with respect to model size, suggesting that larger models have the potential to further improve performance.

## 4.4 PROTEIN-PROTEIN BINDING AFFINITY PREDICTION

**Setup.** For the protein-protein binding affinity task, we adopt the SKEMPIv2 dataset (Jankauskaitė et al., 2019), the most widely used benchmark for $\Delta\Delta\mathcal{G}$ prediction. SKEMPIv2 is a comprehensive, annotated mutation dataset covering 348 protein complexes, containing 7,085 amino acid mutations along with corresponding changes in thermodynamic parameters and kinetic rate constants. Since SKEMPIv2 does not provide structures for the mutated complexes, we generate them using FoldX (Delgado et al., 2019). We evaluate the performance of ADiT using a split-by-complex threefold cross-validation on the SKEMPIv2 dataset following (Liu et al., 2024). The metrics include: (1) Pearson correlation coefficient, (2) Spearman's rank correlation coefficient, (3) root mean squared error (RMSE) and (4) mean absolute error (MAE). Details in Appendix D.4.

Table 4: **Ablation study results on SKEMPIv2 based on ADiT-M.**

| NO. | Model | Pearsonr↑ | Spearmanr↑ | RMSE↓ | MAE↓ |
|-----|-------|-----------|------------|-------|------|
| 0 | **ADiT-M** | 0.683 | 0.539 | 1.559 | 1.098 |
| 1 | w/o pre-training | 0.649 | 0.511 | 1.624 | 1.169 |
| 2 | w/o all-atom info. | 0.658 | 0.517 | 1.606 | 1.153 |
| 3 | w/ larger size | 0.691 | 0.560 | 1.540 | 1.088 |
| 4 | w/ smaller size | 0.660 | 0.524 | 1.597 | 1.132 |

**Results.** The detailed results are presented in Table 2. Among all baselines, ADiT demonstrates strong and consistent performance. The best-performing variant, ADiT-L, achieves the highest Pearson correlation coefficient and the second-best Spearman's rank correlation coefficient. Although this is a ranking task, where metrics like RMSE and MAE are less critical, ADiT still performs comparably to the strongest baselines on these metrics, with only marginal differences.

## 4.5 ANTIBODY-ANTIGEN BINDING AFFINITY PREDICTION

**Setup.** With the models trained on SKEMPIv2, we test their performance on the HER2 binders test set, following (Cai et al., 2024). The HER2 binders test set is collected from (Shanehsazzadeh et al., 2023) and contains high-quality binding affinity data, measured by surface plasmon resonance (SPR) on 419 HER2 binders with de novo designed CDR loops. The antibodies in the dataset are variants of Trastuzumab that have high edit distance (7.6 on average), making them potentially challenging for $\Delta\Delta\mathcal{G}$ bind predictors trained on low-edit-distance data. Here, we use Pearson and Spearman correlation as evaluation metrics to assess the performance of our ADiT. Details in Appendix D.5.

**Results.** Results are shown in Table 3. Our best-performing variant, ADiT-L, demonstrates superior performance, outperforming previous methods Bind-ddG (Shan et al., 2022) and GearBind (Cai et al., 2024). These findings underscore the robust generalization capability of ADiT to held-out antibody data.

**Case Study.** We evaluate ADiT's performance on real-world *in silico* affinity maturation using models trained on SKEMPIv2. Two starting antibodies with different formats and target antigens are tested (see Appendix D.5 for details). Figure 3 shows the rankings of seven wet-lab-validated affinity-enhancing mutations from (Cai

Table 3: **Antibody-antigen binding affinity prediction results.** The best result is highlighted in **bold**, and the second-best result is underlined.

| Model | Pearson↑ | Spear.↑ |
|-------|----------|---------|
| Bind-ddG | 0.387 | 0.388 |
| GearBind | 0.478 | 0.475 |
| GearBind + P | 0.515 | 0.517 |
| FoldX | 0.246 | 0.329 |
| **ADiT-S** | 0.527 | 0.558 |
| **ADiT-M** | 0.507 | 0.523 |
| **ADiT-L** | **0.567** | **0.576** |

et al., 2024)[3]. The results demonstrate that ADiT assigns better ranks to most beneficial mutations (e.g., S54Y and S57W for Anti-5T4 UdAb, and SH103W, SH103Y, and IL34W for CR3022), highlighting its potential as an effective tool for antibody optimization.

## 4.6 ABLATION STUDY

To better understand the key contributors to the performance of ADiT, we conduct an ablation study on the protein-protein binding affinity prediction task using the SKEMPIv2 dataset. We use ADiT-M as the baseline and evaluate the impact of several single-factor modifications: (1) removing pretraining and use randomly initialized parameters, (2) replacing all-atom (AA) structures with backbone-only structures in both the pretraining and finetuning stages, (3) scaling up the model size (35M) to ADiT-L (253M), and (4) scaling down to ADiT-S (12M).

The results are summarized in Table 4. (1) Comparing No.0 and No.1 highlights the importance of pretraining, which significantly contributes to performance gains (5.2% improvement in Pearson, 5.5% improvement in Spearman, 4% improvement in RMSE, 6% improvement in MAE). (2)

---

[3]Only a limited number of mutations were experimentally validated, suggesting that additional affinity-enhancing candidates likely exist.

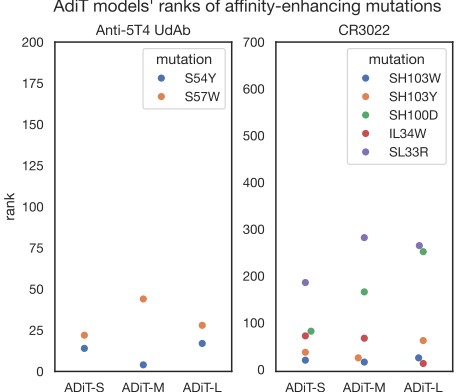

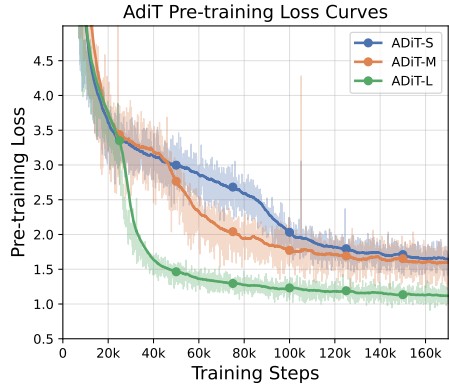

Figure 3: **Case Study on Antibody Affinity Maturation.** ADiT models' ranks of affinity-enhancing mutations for nanobody Anti-5T4 UdAb (780 mutations) and antibody CR3022 (1420 mutations) are shown. Lower rank is better.

Figure 4: **Pre-training and Scaling Effects Across Model Sizes.** Pre-training loss curves for the small, medium, and large variants of ADiT. Larger models consistently achieve lower loss throughout the pre-training process.

The comparison between No.0 and No.2 demonstrates the value of all-atom structural modeling (3.8% improvement in Pearson, 4.3% improvement in Spearman, 3% improvement in RMSE, 4.8% improvement in MAE), which provides richer structural information and leads to improved representations. (3) Finally, the comparison among No.0, No.3, and No.4 reveals consistent performance improvements with increased model size, reflecting a clear scaling trend in line with observations from other domains. Figure 4 presents the pretraining loss curves for the small, medium, and large model variants. Larger models consistently achieve lower loss during pretraining, underscoring their enhanced ability to capture complex molecular patterns.

## 5 CONCLUSION

In this work, we take an important step toward developing all-atom structure foundation models for biomolecular interactions. With large-scale pre-training, our unified foundation model achieves state-of-the-art or competitive performance across diverse tasks. Furthermore, we observe scaling effects in our model architecture that align with trends seen in other domains. Quantitative and qualitative analyses show the practical effectiveness of our approach. We further discuss the LLM usage in Appendix A and limitations in Appendix B.

**Broader Impacts** Our work has significant broader impacts for computational biology and drug discovery. Positively, ADiT provides a powerful, general-purpose framework for modeling biomolecular interactions, potentially accelerating therapeutic discovery and advancing biological understanding. However, ethical concerns, such as the potential misuse in drug development, also warrant careful consideration. Responsible deployment and transparency are essential to maximize benefits while mitigating risks.

## REPRODUCIBILITY STATEMENT

We provide a clear and detailed description of the architecture in Sec. 3, along with the necessary pretraining and finetuning configurations in Sec. D. The pretraining dataset is constructed from publicly accessible PDB and clustered using publicly available clustering results. All finetuning datasets used in our experiments are also publicly accessible. Our open-source implementation is available at `https://github.com/VectorShi/ADiT`.

## ACKNOWLEDGEMENT

The authors acknowledge funding from the Canada CIFAR AI Chair Program and the Intel-Mila partnership program. The computational resources for this project were provided by Mila and the Digital Research Alliance of Canada.

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

## A  LLM Usage Statement

LLMs were used solely for language editing and refinement of the manuscript text. The LLM did not contribute to research ideation, experimental design, data analysis, or content generation beyond linguistic improvements. All scientific concepts, results, and interpretations remain entirely the work of the authors.

The authors reviewed all LLM-suggested changes to ensure accuracy and alignment with the intended meaning, and no text was accepted without human verification.

## B  Limitations.

First, the current model is limited to protein-protein and protein-ligand interactions, excluding other biomolecules such as DNA and RNA. Given AlphaFold 3's demonstrated ability to model a broader range of biomolecular interaction types, extending our approach to encompass all biomolecular interactions is promising for future research. Second, the computational resource constraints have limited our ability to further scale up the model size. We believe that larger models, combined with advanced pre-training objectives, could lead to stronger foundation models. Additionally, while our study provides initial insights into scaling laws for structure foundation models, further exploration with various model sizes is necessary to better understand these trends. Finally, our framework requires fine-tuning for downstream tasks, which poses challenges in scenarios with extremely limited data. Future work should explore zero-shot inference capabilities to broaden the applicability of structure foundation models.

## C  Algorithms

The pseudo codes for "Hierarchical Representation Learning" and "Diffusion Transformer" are shown in Alg 1 and Alg 2.

---

**Algorithm 1** Hierarchical Representation Learning

---

1: **Input:** token condition $c^{\text{token}}$, pair representation $z^{\text{token}}$ and atom condition $c^{\text{atom}}$, single representation $s^{\text{atom}}$ and pair representation $z^{\text{atom}}$
2: **Output:** updated atom single representation $s^{\text{atom}}$
3: # **Step 1: Update atom representations**
4: $s^{\text{atom}} \leftarrow \text{DiffusionTransformer}(s^{\text{atom}}, c^{\text{atom}}, z^{\text{atom}})$
5:
6: # **Step 2: Update token representations**
7: $s^{\text{token}} \leftarrow \text{Atom2Token}(s^{\text{atom}})$
8: $s^{\text{token}} \leftarrow \text{Linear}(c^{\text{token}}) + s^{\text{token}}$
9: $s^{\text{token}} \leftarrow \text{DiffusionTransformer}(s^{\text{token}}, c^{\text{token}}, z^{\text{token}})$
10:
11: # **Step 3: Update atom representations again**
12: $s^{\text{atom}} \leftarrow \text{Token2Atom}(s^{\text{token}}) + s^{\text{atom}}$
13: $s^{\text{atom}} \leftarrow \text{DiffusionTransformer}(s^{\text{atom}}, c^{\text{atom}}, z^{\text{atom}})$
14: **Return** $s^{\text{atom}}$

---

**Algorithm 2** Diffusion Transformer

---

1: **Input:** single representation $s$, pair representation $z$, condition $c$, number of DiT blocks $N_{\text{block}}$
2: **Output:** updated representation $s$
3: **for** $i = 1$ **to** $N_{\text{block}}$ **do**
4:     $s \leftarrow \text{AdaLN}(s, c)$
5:     $s \leftarrow \text{MultiheadAttn}(s, z) + s$
6:     $s \leftarrow \text{Transition}(s, c) + s$
7: **end for**
8: **Return** S

---

# D  EXPERIMENTAL DETAILS

Table 5: Model Hyperparameters.

| Model | ADiT-S | ADiT-M | ADiT-L |
|---|---|---|---|
| **Model Configuration** | | | |
| Atom Repr Dim | 128 | 128 | 128 |
| Atom Pair Dim | 16 | 16 | 16 |
| Token Repr Dim | 384 | 384 | 768 |
| Token Pair Dim | 32 | 64 | 128 |
| Atom Enc Layers | 3 | 3 | 3 |
| Atom Enc Heads | 4 | 4 | 4 |
| Token Attn Layers | 3 | 12 | 24 |
| Token Attn Heads | 8 | 8 | 16 |
| Atom Dec Layers | 3 | 3 | 3 |
| Atom Dec Heads | 4 | 4 | 4 |
| **Pre-Trainig Setup** | | | |
| Optimizer | Adam | Adam | Adam |
| Learning Rate | 1e-4 | 1e-4 | 1e-4 |
| Batch Size per GPU | 6 | 3 | 1 |
| Grad. Acc. Steps | 1 | 2 | 2 |
| Number of GPUs | 4 | 4 | 32 |
| Effective Batch Size | 24 | 24 | 64 |
| Training Steps | 180k | 180k | 180k |
| **Dataset Setup** | | | |
| Truncation Length | 250 | 250 | 200 |
| Random Rotation | yes | yes | yes |

Table 6: Dataset statistics.

| Dataset | Split | Train | Validation | Test | Data Type |
|---|---|---|---|---|---|
| PDB | Random | 142,509 clusters | 7,500 clusters | — | Unlabeled Structure |
| LBA | Identity 30%
Identity 60% | 3,507
3,678 | 466
460 | 490
460 | Protein-Ligand Affinity |
| Davis | Random | 24,044 | 3,005 | 3,005 | Drug-Target Affinity |
| SKEMPIv2 | Fold 0
Fold 1
Fold 2 | 4,765
4,282
4,341 | —
—
— | 1,929
2,412
2,353 | Protein-Protein Affinity |
| HER2 | — | — | — | 419 | Antibody-Antigen Affinity |

All experiments in this paper are performed NVIDIA Tesla A100 GPUs with 40G memory. The dataset statistics are presented in Table 6.

## D.1  PRE-TRAINING CONFIGURATION

We implement and pre-train three variants of ADiT: ADiT-S (12M parameters), ADiT-M (35M parameters), and ADiT-L (253M parameters), using the PDB dataset. Detailed training hyperparameters are listed in Table 5. Notably, ADiT-L adopts the same number of atom- and token-level transformer layers as AlphaFold 3. To accommodate GPU memory constraints, we apply random sequential or spatial cropping to the protein inputs. Additionally, we employ a data augmentation strategy that centers the protein at the origin (based on its center of mass) and applies random rotations. These augmentations help the model learn more robust and generalizable representations.

### D.2 Protein-Ligand Binding Affinity Prediction

**Baselines**  We employ a diverse set of baselines, including B&B (Bepler & Berger, 2019), DeepDTA (Öztürk et al., 2018), DeepAffinity (Karimi et al., 2019), MaSIF (Gainza et al., 2020), Atom3D-GNN (Townshend et al., 2021), IEConv (Hermosilla et al., 2021), Holoprot (Somnath et al., 2021), ProtNet (Wang et al., 2023), Uni-Mol (Zhou et al., 2023), GeoSSL (Liu et al., 2023), ProFSA (Gao et al., 2023), and GET (Kong et al., 2024). Most of the reported results for the baseline methods are taken from (Gao et al., 2023), while we independently evaluate GET (Kong et al., 2024) on the LBA-60 dataset using the same configuration as in the LBA-30 setting reported by the original authors.

To fine-tune Protenix, we modified its Diffusion Module by replacing the noisy coordinates with ground-truth coordinates. We then extracted the output from the last Transformer layer of the AtomAttentionDecoder. This output was passed through a LayerNorm, followed by mean pooling to obtain per-token representations, and then sum pooling to derive a complex-level representation, following the same aggregation strategy used in ADiT. As for the hyperparameters, we kept the architectural settings of Protenix unchanged, since its architecture is fixed. However, we carefully tuned the training hyperparameters to avoid underestimating the baseline performance. Specifically, we used a learning rate of 1e-4, an effective batch size of 16, dropout set to 0.0, and no weight decay.

**Training**  For both the 30% and 60% sequence identity threshold splits, we use a batch size of 16, a fine-tuning learning rate of 1e-4, and apply random structural rotation as data augmentation. For 60% sequence identity threshold split, we additionally apply a dropout rate of 0.5. The total epoches on LBA-30 and LBA-60 are 3 and 99. The training process can be efficiently handled using one or two A100 GPU.

### D.3 Drug-Target Binding Affinity Prediction

**Baselines**  The baselines are: DeepDTA (Öztürk et al., 2018), MT-DTI (Shin et al., 2019), GraphDTA (Nguyen et al., 2021), rzMLP-DTA (Qiu et al., 2021), EnsembleDLM (Kao et al., 2021), MGraphDTA (Yang et al., 2022), FusionDTA (Yuan et al., 2022) and NHGNN-DTA (He et al., 2023). The reported results are taken from (He et al., 2023).

**Training**  We generate unbound protein structures using AlphaFold 2 and molecular structures using RDKit. Since these structures do not initially reside in the same spatial framework, we remove the edges between proteins and ligands to ensure proper alignment. For all five random splits, we use an effective batch size of 8, a learning rate of 1e-4 for ADiT-S and 3e-5 for ADiT-M and ADiT-L, a dropout rate of 0.2 and train for 120 epochs without rotation augmentation. To handle long proteins, we randomly truncate them to a max length of 150. The training process can be efficiently handled using a 4 A100 GPU (for ADiT-L).

### D.4 Protein-Protein Binding Affinity Prediction

**Baselines**  We employ a diverse set of baselines for comparison, including ESM-IF (Hsu et al., 2022), self-attention-based End-to-End network (End-to-End) (Jumper et al., 2021), DDGPred (Shan et al., 2022), MIF-Network (Yang et al., 2020), RDE-Network (Luo et al., 2023), DiffAffinity (Liu et al., 2024), Prompt-DDG (Wu et al., 2024), and Surface-VQMAE (Wu & Li, 2024). The reported results are sourced from (Wu et al., 2024; Wu & Li, 2024).

**Training**  We truncate sequences based on the distance between the current residue and mutation-related residues, limiting the remaining sequence length to 50 residues. Training is conducted with an effective batch size of 8, a learning rate of 3e-5, a dropout rate of 0.0, and a total of 60 epochs. The training process can be efficiently handled using a single A100 GPU.

### D.5 Antibody-Antigen Binding Affinity Prediction

**Baselines**  We compare ADiT against several strong baselines for antibody affinity prediction, including Bind-ddG (Shan et al., 2022), and GearBind (Cai et al., 2024). We also include the physics-

based tool FoldX (Delgado et al., 2019) as an additional baseline. The performance of most baselines is reported from (Cai et al., 2024).

**Case study**    Affinity maturation aims to enhance the binding affinity of an antibody to its target antigen. As the search space for antibodies is too vast for wet-lab enumeration, *in silico* methods are increasingly being used to increase success rates. We can formulate the *in silico* affinity maturation task as a virtual screening problem, where we use a model to rank mutants of the starting antibody by their binding affinity to the target antigen, and validate the top-ranked mutants in wet lab. The two antibodies Anti-5T4 UdAb and CR3022 were chosen in (Cai et al., 2024) because (1) they cover two common antibody formats, nanobody and antibody; (2) they cover two antigens: UdAb targets an oncofetal antigen 5T4, while CR3022 targets SARS-CoV-1 and multiple variants of SARS-CoV-2. The 780 mutations for Anti-5T4 UdAb are obtained from saturation single-point mutagenesis of residues 1, 3, 25, 27-33, 39-45, 52-57, 59, 91-93, 95, 99-103, 105, 110, 112, 115-117 (*i.e.* mutating one of these sites to every possible amino acid type; each site yields 19 mutants), which are residues close to the antibody interface to 5T4. The 1420 mutations for CR3022 are obtained from saturation single-point mutagenesis of residues H25-35, H50-66, H99-108, L24-40, L56-62, L95-103.

## E    ENZYME COMMISSION (EC) NUMBER ANNOTATION TASK

To show ADiT's generalization and applicability to non-binding tasks, we run additional experiments on the Enzyme Commission (EC) number annotation task (Gligorijević et al., 2021), which involves predicting whether a protein can catalyze a chemical reaction. This task is evaluated using the Fmax metric. The results, presented in Table 7, show that our method consistently outperforms the baseline, even under challenging data splits, demonstrating ADiT's applicability beyond binding-related functions.

Table 7: Enzyme Commission (EC) Number Annotation Task.

| METHOD | 30% | 40% | 50% | 70% |
|---|---|---|---|---|
| DEEPFRI | 0.470 | 0.505 | 0.545 | 0.600 |
| GEARNET-EDGE (MULTIVIEW CONTRAST) | 0.744 | 0.769 | **0.808** | **0.848** |
| ADIT-S | **0.751** | **0.773** | 0.806 | 0.843 |

## F    INTERPRETABILITY

Interpretability is a crucial next step toward understanding the mechanisms underlying our model's effectiveness. In this work, due to project scope, we primarily focus on establishing performance and scalability. A comprehensive and systematic interpretability investigation is left for future research. As a preliminary exploration, we conduct a simple case study examining the attention patterns learned by our trained model. We randomly select the `5f63` pocket-ligand complex from the LBA60 test set, which consists of 18 residues in the binding pocket and 41 atoms in the ligand. In this analysis, we extract attention scores from the token-level transformer, since the atom encoder and decoder employ only local block attention.

Several observations arise from this initial investigation:

First, we observe that the average attention weight for intra-chain pairs is **0.0253**, compared to only **0.0056** for inter-chain pairs. This suggests that, although the model captures cross-chain interactions, it places greater emphasis on intra-chain relationships, which may play a more critical role in binding context representation.

Second, the model's attention mechanism appears to effectively "focus" on key interfacing residues and atoms. We visualize the cross-chain attention scores from the second layer (layer index=1, zero-based) in Figure5. To identify potentially important residues and atoms, we highlight the closest one-third of residue tokens to the ligand and the closest one-fifth of ligand atom tokens to the protein. As shown in the results for Head 0 and Head 4, these highlighted residues exhibit higher attention

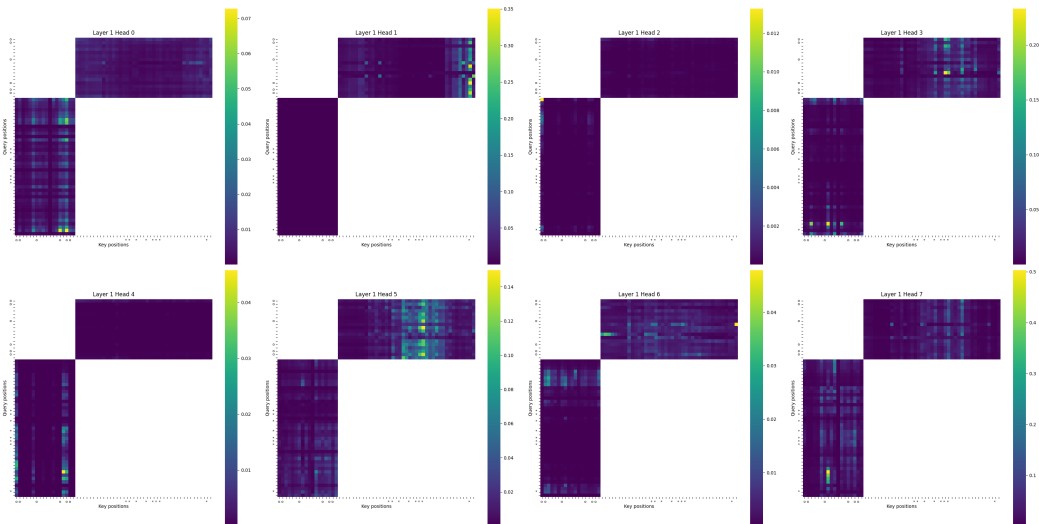

Figure 5: Attention score visualization for the `5f63` pocket-ligand complex, comprising 18 residue tokens and 41 ligand atom tokens. The plot highlights residue-atom interaction patterns. Here, o on the x- or y-axis denotes the closest one-third of residue tokens to the ligand, and ∗ on the x- or y-axis denotes the closest one-fifth of ligand atom tokens to the protein.

weights (columns appear brighter). Similarly, the highlighted ligand atoms receive higher attention in Head 1, Head 3, and Head 5.

