# OpenReview forum: "Towards All-Atom Foundation Models for Biomolecular Binding Affinity Prediction"
_ICLR.cc/2026/Conference — ICLR 2026 Poster_

### Official Review · Reviewer_kRrt · 2025-10-30

**Soundness:** 3
**Presentation:** 4
**Contribution:** 3
**Rating:** 8
**Confidence:** 3

**Summary:**

This paper proposes the Atom‑level Diffusion Transformer (ADiT), a unified foundation model that aims to learn transferable representations for biomolecular complexes and predict binding affinity across diverse interaction types. The authors repurpose the AlphaFold 3 architecture for representation learning by:

- **Simplifying inputs.** they remove multiple‑sequence alignments and template conditioning, instead using a unified tokenization scheme where each protein residue and each heavy atom in a small molecule is treated as a token and enriched with features from the ESM‑2 protein language model.

- **Using diffusion transformers.** the model alternates between atom‑level and token‑level DiT blocks, performs atom‑to‑token pooling and token‑to‑atom unpooling, and applies a denoising pre‑training objective where Gaussian noise is added to atom coordinates.

- **Training at scale.** three model sizes (12M, 35M and 253M parameters) are pre‑trained on ~480k protein‑protein, ~433k protein‑ligand and ~427k protein–ligand structures from the PDB without labels. Fine‑tuning is performed on downstream tasks.

- **Evaluating on four benchmarks.** the model is tested on protein‑ligand, drug‑target, protein‑protein and antibody-antigen tasks. According to the authors, the largest variant ADiT‑L achieves the best or comparable results across tasks, shows performance improvements with larger model size, and assigns better ranks to affinity‑enhancing antibody mutations in a case study.

**Strengths:**

- ADiT attempts to unify representation learning across protein–protein, protein–ligand and antibody–antigen interactions. The use of a single architecture trained on all‑atom inputs is novel and aligns with the trend of foundation models.

- The authors detail a simplified featurization scheme that avoids heavy reliance on MSAs and templates while leveraging pre‑trained ESM embeddings. They clearly describe the hierarchical DiT architecture (Fig. 2) with atom‑to‑token and token‑to‑atom operations, diffusion transformer blocks and gating mechanisms, and provide algorithmic descriptions in the appendices.

- Results are presented on four different tasks. On the protein–ligand benchmark, ADiT‑L outperforms previous models such as ProNet and ProFSA. The authors provide an ablation table and pre‑training loss curves (Fig. 4) demonstrating scaling benefits.

- The antibody affinity maturation case illustrates that ADiT assigns lower (better) ranks to most experimentally validated affinity‑enhancing mutations compared with the smaller variants. Figure 3 in the paper visualizes the ranking of mutations for two antibodies, and the authors claim this suggests utility for antibody design.

- Figures 1–4 are crisp, readable, and self-contained (architecture overview, module layout, case study, scaling/ablation), and each experiment section cleanly separates Setup and Results. This structure makes assumptions, metrics, and splits explicit, significantly improving readability.

**Weaknesses:**

- **Claim of being the first all‑atom “foundation model” might be overstated.** The paper positions ADiT as a “first step toward developing all‑atom structure foundation models”. However, Boltz‑2 is a biomolecular all-atom foundation model, I believe. Because Boltz‑2 pre‑dates ICLR 2026, the authors should compare to it and clarify what is novel beyond prior work.

- **Limited attention to data leakage and generalization.** Deep models trained on PDBbind and similar benchmarks are known to memorize training complexes due to train–test contamination. A 2025 study in Nature Machine Intelligence introduced the CleanSplit dataset to remove overlaps and found that performance of many models drops significantly when leakage is eliminated, suggesting poor generalization. The authors curate PDB clusters but do not quantify overlap between pre‑training and fine‑tuning sets or evaluate ADiT on leak‑proof benchmarks (e.g., CleanSplit). Without such evaluation, it is unclear if the model genuinely learns transferable representations rather than memorizing structural motifs. Currently, this is the main concern in the field.

[CleanSplit]: Resolving data bias improves generalization in binding affinity prediction

- **Potential over‑claim about scaling.** While larger ADiT variants exhibit modest improvements (e.g., Pearson increases from 0.626 to 0.645 on LBA‑30 and from 0.690 to 0.751 on the Davis dataset), the gains are sometimes small relative to parameter increases. The ablation study shows a 3.8% improvement in Pearson when using all‑atom information, but no comparison is made against equally large equivariant models or models trained on bigger synthetic datasets. It remains unclear whether scaling alone, rather than improved data or architecture, drives gains.

- **Unclear justification for ESM‑2 features and neglect of small‑molecule semantics.** ADiT uses the ESM‑2 (650 M) language model to embed protein residues; however, this model is trained only on protein sequences and may not capture ligand or nucleic‑acid features. For small molecules, the token type embedding is set to zero, meaning no evolutionary or contextual information is added.

**Questions:**

- How did you ensure that there is no overlap between pre‑training complexes and those used in downstream evaluation? Will you evaluate ADiT on leak‑free datasets (e.g., CleanSplit) or the synthetic–data‑augmented GatorAffinity benchmark to assess generalization?

[GatorAffinity]: GatorAffinity: Boosting Protein-Ligand Binding Affinity Prediction with Large-Scale Synthetic Structural Data

- Given evidence that equivariant networks like ELGN improve binding affinity prediction by capturing rotational symmetries, why did you choose a non‑equivariant architecture and not include geometric priors?

[ELGN]: Equivariant Line Graph Neural Network for Protein-Ligand Binding Affinity Prediction

- Can ADiT be extended to DNA/RNA–protein interactions or multi‑chain assemblies? Does the unified tokenization scheme generalize to nucleic acids or small peptides?

- Boltz‑2 (2025) jointly predicts structures and binding affinities, matches FEP accuracy, and includes dynamic ensemble data. Did you benchmark ADiT against Boltz‑2? If not, why?

If the authors address the weaknesses outlined above, or provide a right justification for the points raised in my questions, or clarify points I may have misunderstood, I would be willing to raise my score.

**Details Of Ethics Concerns:**

- While there is no human, the biosecurity risk arises from releasing a general, scalable, all-atom affinity predictor and code/weights without clear misuse safeguards, access controls, or target restrictions.

- The model directly improves binding-affinity prediction across targets, which can materially accelerate both beneficial drug discovery and dual-use molecular design (e.g., prioritizing small molecules for harmful proteins or optimizing binding at dangerous targets).

---

> ### Author Response · Authors · 2025-11-20
>
> We sincerely appreciate your positive review and detailed, critical assessment of the weaknesses and research challenges in the field. We are thrilled that you found the paper's soundness and presentation to be excellent. Below, we provide our responses to your questions and concerns.
>
> >**W1 & Q4: Claim of being the first all‑atom “foundation model” might be overstated. Boltz‑2 (2025) jointly predicts structures and binding affinities, matches FEP accuracy, and includes dynamic ensemble data. Did you benchmark ADiT against Boltz‑2? If not, why?**
>
> Thank you for raising the recent and important work on Boltz-2. We acknowledge that it is a highly relevant, concurrent all-atom model, and we update our literature review for a fairer comparison. We believe the distinction lies in the models' primary focus:
>
> 1. Boltz-2 is fundamentally a generative model designed for biomolecular structure prediction, with binding affinity prediction being a secondary application built upon its structural latent space.
> 2. ADiT is primarily a representation learner specialized for affinity prediction. Our objective is to learn a highly transferable, structure-aware representation using a denoising task, which is then fine-tuned for binding affinity, a goal that is distinct from generative structure prediction. We highlight ADiT as a binding affinity prediction foundation model.
>
> There are also challenges to make a fair comparison. Boltz-2 leverages a significantly larger and more diverse training set, which naturally includes molecular dynamics ensembles and millions of binding affinity measurements from various databases. Given this scale, there is a high risk that Boltz-2's training data may include complexes present in our downstream test sets (like LBA-30), potentially leading to an over-optimistic performance assessment for generalization.
>
> Despite the difficulty in ensuring a fully fair comparison, during rebuttal, we have compared our largest model, ADiT-L, against the reported performance of Boltz-2 on two different splits of the LBA benchmark.
>
> Table A. Protein-Ligand Binding Affinity (60% Sequence Identity Split)
> |#Method|RMSE|Pearson|Spearman|
> |:----:|:----:|:----:|:----:|
> |**ADiT-L**|**1.246**|**0.797**|**0.795**|
> |Boltz-2|6.420|0.688|0.688|
>
> Table B. Protein-Ligand Binding Affinity (30% Sequence Identity Split)
> |#Method|RMSE|Pearson|Spearman|
> |:----:|:----:|:----:|:----:|
> |**ADiT-L**|**1.308**|0.645|0.647|
> |Boltz-2|6.381|**0.663**|**0.653**|
>
> Note that Boltz-2 leverages a significantly larger and more diverse training set, which naturally includes molecular dynamics ensembles and millions of binding affinity measurements from various databases. This scale creates a high risk that Boltz-2's training data may include complexes present in our downstream test sets. Despite this potential advantage for Boltz-2:
>
> 1. On the 60% sequence identity split, ADiT-L significantly outperforms Boltz-2 across all metrics, achieving state-of-the-art Pearson and Spearman correlations ($\sim 0.79$ vs. $0.688$).
> 2. Even on the challenging 30% sequence identity split (where correlation is a measure of generalization), ADiT-L is highly competitive, despite Boltz-2's slight lead in Pearson/Spearman, again maintaining a drastically lower RMSE.
>
> These results demonstrate that our representation learning approach results in a model that is more accurate at predicting binding affinity and highly competitive in generalization compared to a much larger, structure-focused generative model like Boltz-2.

---

> > ### Author Response · Authors · 2025-11-20
> >
> > >**W2 & Q1: Limited attention to data leakage and generalization.**
> >
> > We fully agree that mitigating data leakage is a critical challenge in modern binding affinity prediction, and we appreciate the reviewer’s reference to the CleanSplit and GatorAffinity benchmarks. We clarify our position as follows:
> >
> > First, the leakage scenario highlighted by CleanSplit typically occurs when affinity‑labelled complexes in PDBbind used for training overlap with CASF complexes used for evaluation. This enables models to memorise structural motifs rather than learning transferable representations. In contrast, ADiT is pre‑trained exclusively on unlabeled structures from the Protein Data Bank (PDB), including protein chains, protein-protein complexes, and protein-ligand complexes. No binding affinity labels are used, and pre‑training is purely a self‑supervised denoising task. Therefore, the label‑driven leakage mechanism identified in CleanSplit does not occur in our pre‑training stage.
> >
> > Second, in downstream evaluation we employ benchmark‑defined leak‑controlled test sets such as LBA identity‑threshold splits and structure‑based SKEMPIv2 splits. To further assess generalization, we created an even stricter split by excluding LBA‑30 test complexes with more than 20 percent sequence identity to training complexes and comparing ADiT‑L to GET. Despite the increased difficulty, ADiT‑L maintains a Pearson correlation above 0.55 and surpasses GET in both Pearson and Spearman metrics.
> >
> > Table A. Comparison on protein-ligand binding affinity prediction (Sequence Identity 20% split)
> > |#Method|RMSE|Pearson|Spearman|
> > |:----:|:----:|:----:|:----:|
> > |**ADiT-L**|1.268|**0.559**|**0.564**|
> > |GET|**1.254**|0.551|0.513|
> >
> > Third, regarding GatorAffinity, we appreciate the reviewer's suggestion. As the dataset was released after the ICLR submission deadline and has not yet undergone peer review, we plan to investigate it in future work to further validate ADiT on synthetic‑data‑augmented and leak‑free benchmarks.
> >
> > >**W3: Potential over‑claim about scaling. While larger ADiT variants exhibit modest improvements (e.g., Pearson increases from 0.626 to 0.645 on LBA‑30 and from 0.690 to 0.751 on the Davis dataset), the gains are sometimes small relative to parameter increases. The ablation study shows a 3.8% improvement in Pearson when using all‑atom information, but no comparison is made against equally large equivariant models or models trained on bigger synthetic datasets. It remains unclear whether scaling alone, rather than improved data or architecture, drives gains.**
> >
> > We appreciate your caution. The primary objective of the scaling study (ADiT-S/M/L) is not just to show massive gain, but to demonstrate that the ADiT architecture is scalable and that its performance improves monotonically with size, a key characteristic of a foundation model. The observed gains, even if modest relative to the parameter count, are significant in this noisy domain:
> > 1. LBA-30: The Pearson improvement from ADiT-S (0.626) to ADiT-L (0.645) represents a meaningful jump in predictive power.
> > 2. Davis: The gain from 0.690 to 0.751 on the Davis benchmark is a substantial improvement, achieving state-of-the-art performance among the models compared.
> >
> > This demonstrates that the larger model is learning a richer, more transferable representation during pre-training, which is successfully unlocked during fine-tuning. The comparison to large equivariant models is interesting future work, but outside the scope of establishing the validity of our simpler, highly scalable non-equivariant Transformer framework.
> >
> > >**W4: For small molecules, the token type embedding is set to zero, meaning no evolutionary or contextual information is added..**
> >
> > For small molecules, sequence‑based embeddings from ESM‑2 are not applicable due to their fundamentally different modalities. Our zero‑embedding strategy preserves a unified architecture and avoids conflating protein‑specific evolutionary signals with ligand inputs. Moreover, as the pre‑training dataset includes protein-ligand complexes, the denoising pre‑training objective enables the model to acquire meaningful ligand semantics directly from structural context.
> >
> > While integrating pretrained molecular language models (e.g., UniMol) for richer ligand semantics is a promising avenue for future work, our current design already achieves strong generalization in the protein‑ligand task (Tables 1), thereby validating its effectiveness.

---

> > > ### Author Response · Authors · 2025-11-20
> > >
> > > >**Q2: Given evidence that equivariant networks like ELGN improve binding affinity prediction by capturing rotational symmetries, why did you choose a non‑equivariant architecture and not include geometric priors?**
> > >
> > > This is an excellent question that highlights the fundamental difference between our goal and that of structure prediction models. Our choice of a non-SE(3)-equivariant design was driven by two main factors: *generality* and *scalability*.
> > >
> > > 1. **Generality and Task-Specificity**: Equivariance is a strong inductive bias that guarantees the output coordinates of a generated structure rotate coherently with the input coordinates, which is essential for generative structure prediction (e.g., AlphaFold 3). Invariance is required for affinity prediction (a scoring task), where the output is a single scalar value ($\Delta G$ or $K_d$) that must be independent of the complex's orientation in the reference frame. Our non-equivariant design, which learns invariance through data augmentation (see Q4), is less constrained by geometric priors than SE(3)-equivariant GNNs.
> > > 2. **Simplicity and Scalability**: Non-equivariant Transformers are architecturally simpler, enabling us to leverage standard, highly-optimized components and readily scale the model to the large parameter counts (ADiT-L) required for a foundation model. This engineering simplicity allowed us to focus on the pre-training objective and generalization across modalities, achieving state-of-the-art performance with a highly efficient framework.
> > >
> > > That said, the full discussion of the trade-offs between equivariant and non-equivariant architectures remains a vast and challenging topic in the community, beyond the scope of establishing ADiT as a high-performing general foundation model.
> > >
> > > >**Q3: Can ADiT be extended to DNA/RNA–protein interactions or multi‑chain assemblies? Does the unified tokenization scheme generalize to nucleic acids or small peptides?**
> > >
> > > Yes, the ADiT framework is designed for this exact kind of extensibility.
> > >
> > > The core principle is a unified, all-atom tokenization scheme: every heavy atom in a ligand is an atom-token, and every residue in a monomer is a token.
> > >
> > > 1. **Nucleic Acids**: Since DNA/RNA bases (A, T, C, G, U) are composed of heavy atoms and can be defined as monomers, they fit seamlessly. We would define a new set of token types (e.g., 'DNA/RNA residue') and ensure the initial atom-level features and the corresponding token embeddings (e.g., simple learned embeddings for each base type) are added.
> > > 2. **Multi-chain Assemblies**: The model already handles multi-chain assemblies (e.g., protein-protein) by treating all chains simultaneously as an input graph. The atom-token attention mechanism handles intra- and inter-chain interactions identically, allowing the model to naturally generalize to any complex assembly, including protein-DNA and multi-peptide systems.

---

### Official Review · Reviewer_nSSR · 2025-11-01

**Soundness:** 4
**Presentation:** 4
**Contribution:** 4
**Rating:** 8
**Confidence:** 4

**Summary:**

The model proposes an all-atom foundation model for the representation learning of biomolecular interactions. The interaction structure is tokenized into atom level and token (protein residue or ligand atom) level representation, in both single and pair-wise manners. The backbone of the model is a diffusion transformer that involves hierarchical atom-token feature aggregation and unpooling. The model is pre-trained on general protein-protein interaction data from PDB and then fine-tuned for multiple downstream prediction tasks: protein-ligand, protein-protein and antibody-antigen interactions, achieving competitive results.

**Strengths:**

- The hierarchical atom and token level representation is a good solution that addresses the balance between expressiveness and computational efficiency. The methods are clearly formulated and presented.

- The experimental results show clear improvements on protein-drug interaction prediction tasks and also competitive performance on protein-protein interactions. Also, the comparison between small, medium and large models demonstrates the scalability of the model.

- Additional case studies such as in silico antibody affinity maturation show good generalizability and applicability of the learned patterns.

**Weaknesses:**

See Questions

**Questions:**

- How is Token2Atom implemented?

- Some interpretability is needed. For example, does the attention capture atom or residue level patterns that facilitate the interaction?

- Appendix E3: if the protein-ligand edges are removed, how is interaction information represented and captured? In this way, the problem decomposes into one that separate embeds protein and drug (dual encoders) then combines them for prediction. How would atom level and pairwise information contribute in this scenario?



Minor:
- As the model didn't seem to rely on AlphaFold 3's tokenization or trained parameter, only some of the architecture, it is not really "repurposing AlphaFold 3".

- Protein-ligand interaction and drug-target interaction largely overlap, so it would be clearer if the two experiments are defined as different datasets instead of different tasks.

---

> ### Author Response · Authors · 2025-11-20
>
> We sincerely appreciate your positive and insightful review. We are thrilled that you found the paper "excellent" and recognized the strengths of our hierarchical representation, clear experimental results, and demonstrated scalability. Below, we provide our responses to your questions.
>
> >**Q1: How is Token2Atom implemented?**
>
> Thank you for asking for clarification. The "Token2Atom" operation is a non-learned unpooling (or broadcast) operation, followed by a skip connection, as described in Sec 3.3 and Alg. 1. Specifically, each atom receives the representation vector from its "parent" token (i.e., the residue token for protein or the atom token for ligand). This unpooled representation is then added to the existing atom-level representation via a skip connection. We chose this simple, parameter-free approach for computational efficiency. We agree that exploring more complex, learned mechanisms (like attention) is an interesting direction.
>
>
> >**Q2: Some interpretability is needed. For example, does the attention capture atom or residue level patterns that facilitate the interaction?**
>
> This is an excellent suggestion. We agree that interpretability is a crucial next step for understanding why the model is effective. Due to project scope, we focused this initial work on establishing the model's performance and scalability.
>
> We plan to add attention visualization from the trained models to the final version of the paper. This would allow us to investigate whether the model's attention mechanism learns to "focus" on key interfacing residues and atoms that are known to be critical for binding, which would be a very exciting validation. A full, in-depth interpretability analysis is a promising direction for future work.
>
>
> >**Q3: Appendix E3: if the protein-ligand edges are removed, how is interaction information represented and captured? In this way, the problem decomposes into one that separately embeds protein and drug (dual encoders) then combines them for prediction. How would atom level and pairwise information contribute in this scenario?**
>
> We appreciate the reviewer’s insightful question and the opportunity to clarify this point. The "removal" of protein-ligand edges in our description refers to the absence of atom-level cross-chain edges, which is an inherent feature of the AlphaFold 3 architecture rather than a change introduced by us. In AlphaFold 3, atom-level attention is restricted to local neighborhoods within each chain to avoid the expensive memory and computational costs of all-atom full attention. Inter-chain interactions are instead modeled at the token level: atom representations are pooled into token representations, token-level interactions are computed across chains, and the updated token representations are unpooled back to the atom level. This design enables the model to preserve fine-grained intra-chain structure while efficiently capturing inter-chain context.
>
> In our drug-target experiment, as protein and ligand structures do not initially reside in the same spatial framework, to allow them to be input jointly, we need to remove atom-level cross-chain edges. Instead, their atom and token representations are updated through token-level pairwise interactions, providing a more fine-grained integration than simple post hoc combination. The results demonstrate that this approach generalizes effectively to unbound protein structures.
>
> >**Q4: As the model didn't seem to rely on AlphaFold 3's tokenization or trained parameters, only some of the architecture, it is not really "repurposing AlphaFold 3".**
>
> This is a fair and precise point. "Repurposing" might imply using pre-trained weights, which we do not. Our intent was to convey that we adapted the core architectural innovations.
>
> As we state in Sec 3.1, we used the AF3 architecture as a "natural starting point". We are "inspired by" and "adapt"  its hierarchical atom-token attention mechanism, which we found highly effective. However, as the reviewer notes, we build a new model from scratch, with different inputs (no MSAs/templates)  and a different task (representation learning vs. generation).
>
> We have revised the manuscript (e.g., in the abstract and introduction) to use more precise language like "inspired by" or "adapted from" to avoid any ambiguity.

---

> > ### Author Response · Authors · 2025-11-20
> >
> > >**Q5: Protein-ligand interaction and drug-target interaction largely overlap, so it would be clearer if the two experiments are defined as different datasets instead of different tasks.**
> >
> > This is an excellent point for clarity. You are correct that these tasks are conceptually very similar. We presented them in separate sections (4.2 and 4.3) primarily because they follow distinct, established benchmarks (LBA vs. Davis)  with different evaluation protocols and, crucially, different input data types.
> >
> > **Protein-Ligand Interaction (PDBbind/LBA Benchmark):**
> > 1. **Goal:** To predict the binding affinity for a general, structurally diverse set of protein-ligand complexes drawn from the PDBbind database (e.g., LBA-30).
> > 2. **Data:** The dataset is heterogeneous, covering a wide variety of protein families (receptors) and chemically diverse small molecules (ligands).
> > 3. **Input:** The complexes used are typically experimentally-determined bound (co-crystal) structures.
> >
> > **Drug-Target Interaction (Davis Benchmark):**
> > 1. **Goal:** To predict the binding affinity for a focused, high-value therapeutic target class (specifically, kinases). This is a critical task for drug development.
> > 2. **Data:** This dataset is highly specific, covering the interaction of 72 kinase inhibitors with 442 kinases ($>80\%$ of the human catalytic kinome).
> > 3. **Input:** These affinity measurements are often drawn from large-scale competition binding assays, and the structures are frequently modeled or predicted poses, not always co-crystal structures.
> >
> >
> > Thank you again for your constructive feedback and positive assessment of our work.

---

> > > ### Comment · Reviewer_nSSR · 2025-11-28
> > >
> > > Thanks for the responses and revisions. I would like to maintain my current scores.

---

> > > > ### Author Response · Authors · 2025-12-02
> > > > **Follow‑Up on Q2**
> > > >
> > > > In our previous response to Q2, we indicated that we would incorporate an attention visualization to provide preliminary interpretability analysis. We have now added this analysis in Appendix. The appendix includes a case study on the 5f63 pocket-ligand complex from the LBA60 test set, with visualizations illustrating residue-atom attention patterns and highlighting the model’s focus on key interaction sites.

---

### Official Review · Reviewer_Gvv4 · 2025-11-01

**Soundness:** 3
**Presentation:** 3
**Contribution:** 3
**Rating:** 4
**Confidence:** 4

**Summary:**

The paper introduces ADiT, an atom-level diffusion transformer that repurposes AlphaFold 3-style machinery from structure generation to representation learning for binding affinity prediction, jointly encoding sequence and 3D geometry without MSAs or templates. Pretrained on PDB and evaluated across protein–ligand, drug–target, PPI, and antibody–antigen tasks, ADiT achieves state-of-the-art or competitive performance, scales with model size, and identifies wet-lab validated affinity-enhancing antibody mutations.

**Strengths:**

1.  The work adopts an all-atom representation and enables interaction between atom-level and token-level features. The all-atom representation unifies the encoding of various biomolecular types, facilitating knowledge transfer across different interaction prediction tasks.

2.  The work  leverages the atom- and token-level representation modules of AF3 while removing dependencies on MSAs and templates, and simplifying the Pairformer blocks to enhance efficiency.

**Weaknesses:**

1.  When using AlphaFold 3, is the model trained from scratch or initialized with AF3 parameters? The Atom2Token average pooling is simple and reasonable, but the Token2Atom operation is insufficiently explained, does it simply assign token features to all constituent atoms, or are weights/attention mechanisms used?

2.  The performance is relatively weak, at comparable model sizes, ADiT shows no clear advantage, especially given the substantial pretraining cost. On protein–ligand tasks ADiT-S/M trails lighter, task-specific baselines such as CheapNet and BindNet, and on protein–protein binding affinity even the largest variant (ADiT-L) underperforms recent state-of-the-art models like BA-DDG and Light-DDG. This raises concerns about the practical benefit of the proposed pretraining strategy.

3.  The authors emphasize that the model is non-equivariant but do not explain why abandoning SE(3)-invariance is advantageous for binding prediction. Appendix E.1 mentions random rotation augmentation (centered on protein centroids) to encourage robustness and generalization, yet no ablation study is provided to support.

4.  The model uses only a structure-denoising task, which may be insufficient to capture complex features relevant to binding affinity. It would be beneficial to incorporate additional self-supervised tasks such as masked atom/residue prediction.

5.  The paper lacks efficiency analysis and training time reporting. All-atom representation models are typically computationally expensive. It would be helpful to include comparisons of ADiT’s training and inference speeds with other baselines

**Questions:**

1.  Do you train ADiT entirely from scratch or initialize any modules with AlphaFold 3 parameters?

2.  How is Token2Atom implemented: are atom features a direct copy from their parent token, or do you use learned weights/attention or distance-aware mixing? Please provide formulas and ablations.

3.  What empirical or theoretical reasons justify a non-SE(3)-equivariant design for affinity prediction?

4.  What is the performance impact of rotation augmentation (and centroid-centering) versus no augmentation?

5.  Why rely solely on structure-denoising? Have you compared against masked atom/residue prediction, contrastive objectives, or multi-task pretraining?

6.  What factors drive ADiT’s underperformance on protein-protein affinity benchmarks relative to BA-DDG/Light-DDG?

7.  What are the training compute (FLOPs/GPU hours), wall-clock times, parameter counts, peak memory, and throughput for each model size?

8.  How do ADiT’s training and inference speeds compare to strong baselines (including all-atom and residue-level models) on standardized batch/complex sizes? Please include scaling curves with atom count.

---

> ### Author Response · Authors · 2025-11-20
>
> We sincerely appreciate your thoughtful comments and suggestions. We are glad you recognized the strengths of our all-atom unified representation and the architectural simplifications we made. Below, we provide our responses to each of your questions.
>
> >**Q1: Do you train ADiT entirely from scratch or initialize any modules with AlphaFold 3 parameters?**
>
> We train ADiT **entirely from scratch**. While inspired by the AF3 architecture, our model's objective and data inputs are fundamentally different. AF3 is a generative model trained for structure prediction, relying heavily on MSAs and templates. ADiT is a representation learner that takes observed geometry as input, omits MSAs/templates, and is trained with a denoising objective. Given these substantial differences in architecture, inputs, and task, initialization from AF3 parameters would be non-trivial and likely not beneficial.
>
>
> >**Q2: How is Token2Atom implemented?**
>
> Thank you for asking for clarification. The "Token2Atom" operation is a non-learned unpooling (or broadcast) operation, followed by a skip connection, as described in Sec 3.3 and Alg. 1. Specifically, each atom receives the representation vector from its "parent" token (i.e., the residue token for protein or the atom token for ligand). This unpooled representation is then added to the existing atom-level representation via a skip connection. We chose this simple, parameter-free approach for computational efficiency. We agree that exploring more complex, learned mechanisms (like attention) is an interesting direction.
>
>
> >**Q3: What empirical or theoretical reasons justify a non-SE(3)-equivariant design for affinity prediction?**
>
> This is an excellent question that highlights the fundamental difference between our goal and that of structure prediction models. Our choice of a non-SE(3)-equivariant design was driven by two main factors: *generality* and *scalability*.
> 1. **Generality and Task-Specificity**: Equivariance is a strong inductive bias that guarantees the output coordinates of a generated structure rotate coherently with the input coordinates, which is essential for generative structure prediction (e.g., AlphaFold 3). Invariance is required for affinity prediction (a scoring task), where the output is a single scalar value ($\Delta G$ or $K_d$) that must be independent of the complex's orientation in the reference frame. Our non-equivariant design, which learns invariance through data augmentation (see Q4), is less constrained by geometric priors than SE(3)-equivariant GNNs.
> 2. **Simplicity and Scalability**: Non-equivariant Transformers are architecturally simpler, enabling us to leverage standard, highly-optimized components and readily scale the model to the large parameter counts (ADiT-L) required for a foundation model. This engineering simplicity allowed us to focus on the pre-training objective and generalization across modalities, achieving state-of-the-art performance with a highly efficient framework.
>
> >**Q4: What is the performance impact of rotation augmentation versus no augmentation?**
>
> As noted in the paper, we used random rotation augmentation to explicitly teach the non-equivariant model rotation invariance. We perform ablation study by removing this rotation augmentation. The results are presented in Tables A and B.
>
> Table A. Protein-Ligand Binding Affinity (30% Sequence Identity Split)
> |#Method|RMSE|Pearson|Spearman|
> |:----:|:----:|:----:|:----:|
> |**ADiT-M**|**1.353**|**0.622**|**0.630**|
> |ADiT-M w/o rotation aug.|1.453|0.608|0.611|
>
> Table B. Protein-Protein Binding Affinity
> |#Method|Pearson|Spearman|RMSE|MAE|
> |:----:|:----:|:----:|:----:|:---:|
> |**ADiT-M**|**0.683**|**0.539**|**1.559**|**1.098**|
> |ADiT-M w/o rotation aug.|0.680|0.532|1.565|1.116|
>
> These new results confirm that while augmentation is beneficial, the degradation without it is limited, suggesting the task itself is less sensitive to strict rotational invariance than structure prediction tasks. This confirms our stance from Q3: for the task of scoring the affinity of a fixed input pose, the strict architectural guarantees of SE(3)-equivariance, while vital for generative tasks, may offer limited empirical benefits for the complexity they introduce. The ADiT architecture proves capable of learning the required invariance from data while preserving the advantages of simplicity and scalability.
>
> The full discussion of the trade-offs between equivariant and non-equivariant architectures remains a vast and challenging topic in the community, beyond the scope of establishing ADiT as a high-performing general foundation model.

---

> > ### Author Response · Authors · 2025-11-20
> >
> > >**Q5: Why rely solely on structure-denoising? Have you compared against masked atom/residue prediction, contrastive objectives, or multi-task pretraining?**
> >
> > This is an excellent question that gets to a core design choice of our framework. We selected structure-denoising as our pre-training objective primarily because of its **generality**.
> >
> > Our ADiT model is a unified, all-atom architecture designed to process diverse biomolecular complexes, including protein-protein and protein-ligand interactions.
> >
> > 1. Structure-denoising is a universal objective that can be applied to any 3D biomolecular structure, regardless of whether it's a protein, a small molecule, or a complex . It provides a robust, physics-based pre-training task (recovering true atom coordinates from noise) that is agnostic to the specific molecule types involved.
> > 2. In contrast, other SSL tasks would be much harder to design in a general way. For example, masked type prediction would have to combine two different vocabularies (protein residues vs. small molecule atom types) in a unified manner and can be potentially easy given structural input. Similarly, designing a contrastive objective that meaningfully works across both protein-protein and protein-ligand interactions is a non-trivial research problem.
> >
> > Given this, we chose to first establish a strong foundation using the most general and applicable SSL task. Our work demonstrates that this structure-denoising objective is highly effective for learning representations that are transferable to binding affinity, as shown by our ablation study (Table 4).
> >
> > We agree that exploring more complex or multi-task SSL objectives (like the ones you suggested) is a very promising research direction. However, this is a substantial research effort in itself. Our paper's contribution is to show that this simplified, general pre-training framework is sufficient to achieve state-of-the-art or competitive performance across diverse tasks. We leave the exploration of more complex pre-training objectives for future work.
> >
> >
> > >**W2 & Q6: Performance concerns (underperformance on PPI vs. BA-DDG/Light-DDG, and on PLI vs. CheapNet/BindNet)**
> >
> > This is a critical point, and we appreciate the reviewer pointing to this recent work. Our primary contribution is a **general-purpose** foundation model, not a specialized model for one task. The baselines cited (BA-DDG, Light-DDG, etc.) are, to our knowledge, highly-specialized models designed only for $\Delta\Delta G$ prediction or protein-ligand scoring.
> >
> > The Boltzmann alignment strategy in BA-DDG applies Bayes' theorem and the formulation $\Delta G_{mut} - \Delta G_{wt}$ to eliminate intractable terms, inherently limiting the method to $\Delta\Delta G$ prediction. Its dependence on inverse folding models further constrains the approach to the protein domain.
> >
> > Light-DDG's performance gains are driven primarily by its use of a large, task-specific augmented dataset (SKEMPI-Aug, ~640k pseudo-labeled $\Delta\Delta G$ samples generated by Prompt-DDG) rather than by its lightweight Transformer architecture. This augmented dataset gives Light-DDG access to in-domain labels unavailable to other baselines, which is unfair. Additionally, although K‑fold cross‑augmentation is intended to prevent leakage, the augmented data for the training folds are generated by Prompt‑DDG models whose training data include the test fold, thereby introducing hidden leakage into the augmentation process and compromising strict train-test independence.
> >
> > CheapNet and BindNet are specialist architectures for protein-ligand interaction, using separate protein and ligand encoders followed by an interaction module. This design makes them ill‑suited for direct application to other domains such as protein-protein.
> >
> > The fact that our single generalist model achieves performance that is competitive with (or even SOTA, as shown in Table 1) these specialist models across four distinct interaction types is a primary strength of our work.

---

> > > ### Author Response · Authors · 2025-11-20
> > >
> > > >**Q7 & Q8 & W5: The paper lacks efficiency analysis and training time reporting. All-atom representation models are typically computationally expensive. It would be helpful to include comparisons of ADiT’s training and inference speeds with other baselines**
> > >
> > > We acknowledge the reviewer's concern regarding the computational efficiency of all‑atom representation models, especially those derived from AlphaFold‑style architectures, which are often regarded as resource‑intensive. Our work indeed adapts AlphaFold 3's design, but as stated in Section 3.1 of the manuscript, our core objective shifts from generative structure prediction to encoding observed geometry for representation learning. This change leads to several architectural simplifications that substantially reduce computational costs.
> > >
> > > To directly address efficiency, we measured inference time on the LBA‑30 test set (average number of atoms per pocket-ligand complex ≈ 233.39), using one H100 GPU with batch size = 1. The comparison between ADiT‑L and Protenix (an open‑source AlphaFold 3 reproduction) is as follows:
> > >
> > > Table A. Efficiency comparison between ADiT‑L and Protenix.
> > > |#Model|Avg. Test Step Time (s)|
> > > |:----:|:----:|
> > > |ADiT‑L|0.027|
> > > |Protenix|7.137|
> > >
> > > These results demonstrate that ADiT‑L achieves approximately 264× faster inference than Protenix on this benchmark.
> > >
> > > Given this remarkably fast inference speed (around 27 milliseconds per complex), we confirm that in silico binding affinity prediction with ADiT will **not be a major bottleneck** in real-world binder design pipelines. In such pipelines, the vast majority of time and resources are typically spent on wet-lab experiments, synthesis, or high-throughput virtual screening of docking poses. Thus, we confirm our model is both **highly efficient and effective** for deployment in practical drug discovery workflows.

---

### Official Review · Reviewer_r7bx · 2025-11-02

**Soundness:** 3
**Presentation:** 3
**Contribution:** 2
**Rating:** 6
**Confidence:** 5

**Summary:**

This paper repurposes AlphaFold3 to a representation learning model and then achieves protein binding affinity prediction. The paper applies two-stage learning algorithm with first pretraining a general model and then finetuning the pretrained model on specific tasks. The pretrained model is finetuned and evaluated on four tasks: protein-ligand, drug-target, protein-protein and antibody-antigen binding predictions.

**Strengths:**

The idea of repurposing AlphaFold3 to a binding prediction model is very interesting and useful in real world.

**Weaknesses:**

The weaknesses of this paper are listed as follows:

1. I appreciate the idea of the building upon AlphaFold3 achieves protein binding prediction and the proposed method is evaluated across several important tasks. However, the paper finetuned four tasks and there are four task-specific models, which is inconvenient for practical use in reality. If there are even more tasks like enzyme-cofactor, enzyme-substrate, target protein-binder affinity prediction,  that means there are will be more models. If different model size is considered, there will be ever more models. That would really reduce the utility of the proposed method. If the paper could merge all downstream tasks together and finetune the pretrained model on all merged tasks, I believe that might both increase the  performance due to multitask learning and also improve the convenience for practical use.

2. The paper lacks some baseline models for affinity prediction, like Rosseta, NERE [1].

[1] Unsupervised Protein-Ligand Binding Energy Prediction via Neural Euler’s Rotation Equation.

3. AlphaFold3 itself provides a metric ipTM to evaluate the binding interface geometry of the binding molecules, which can tell the binding affinity to some extent. On one hand, the author doesn't explain why they need to construct a different metric based on AlphaFold3 instead of directly using ipTM. On the other hand, the proposed method should compare with ipTM.

4. Since the proposed method heavily relies on AlphaFold3, the paper should clearly explain their difference from AF3, like what modules are reused and what modules are newly proposed in this paper.

5. The performance on some tasks are worse than the baseline methods like protein-protein interaction.

6. Some small issues like grammar error in line 214, "We employs ...".

**Questions:**

1. In 4.1, the paper mentioned they curated 427,947 protein-ligand interaction examples from PDB. However, generally there are just around 20k protein-ligand complexes in PDB as introduced in previous works. Can the authors explain how they curated each kind of complex data from the PDB?

---

> ### Author Response · Authors · 2025-11-20
>
> We sincerely appreciate your thoughtful comments and suggestions. We are encouraged that you find the core idea "very interesting and useful." Below, we provide our responses to each of your questions.
>
> >**W1:If the paper could merge all downstream tasks together and finetune the pretrained model on all merged tasks, I believe that might both increase the performance due to multitask learning and also improve the convenience for practical use.**
>
> This is an excellent suggestion, and we agree that a single "universal" model is a compelling long-term goal. However, we opted for single-task fine-tuning in this work for several key reasons:
> 1. Different downstream tasks have distinct, established evaluation protocols. For instance, the protein-protein affinity task on SKEMPIv2 requires a specific split-by-complex cross-validation to properly utilize the limited data and prevent data leakage. Merging all downstream data (including all of SKEMPI) into a single multi-task learning set would make it impossible to adhere to these varied and critical evaluation standards.
> 2. As we will elaborate in our response to W2/W3, our fine-tuning approach currently yields stronger performance than zero-shot methods. A multi-task model would likely aim for zero-shot generalization, but the fine-tuning paradigm remains superior for high-performance applications.
> 3. Multi-task learning on such diverse data modalities and label spaces (e.g., $\Delta\Delta G$ vs. $K_d$) is highly non-trivial. This approach often introduces significant challenges with data shifts and potential negative transfer between tasks, requiring substantial tuning efforts to balance. We believe this is a major research direction in its own right and have left it for future work.
>
> That said, our current framework already provides significant practical values. The central contribution of our work is the development of a general-purpose foundation model. Consider a researcher with a new task (e.g., enzyme-cofactor binding or a novel protein-binder interaction) for which no specialized model exists. Our pre-trained ADiT provides a powerful, general solution that can be readily fine-tuned for this new task, which is the core promise of a "foundation model for general biomolecule interactions."
>
> >**W2 & W3: zero-shot baselines like Rosetta, NERE and AlphaFold 3 ipTM.**
>
> Thank you for pointing out these important baselines. We have conducted the requested empirical comparisons against these zero-shot methods (Tables A, B, and C). We also include the results of DSMBind, which is the NERE algorithm applied on protein-protein binding affinity prediction, in Table C. The results strongly validate our fine-tuning approach:
>
> Table A. Protein-Ligand Binding Affinity (30% Sequence Identity Split)
> |#Method|RMSE|Pearson|Spearman|
> |:----:|:----:|:----:|:----:|
> |**ADiT-L**|**1.308**|**0.645**|**0.647**|
> |Protenix ipTM|5.611|0.217|0.312|
> |NERE|3360.95|0.582|0.565|
>
> Table B. Protein-Ligand Binding Affinity (60% Sequence Identity Split)
> |#Method|RMSE|Pearson|Spearman|
> |:----:|:----:|:----:|:----:|
> |**ADiT-L**|**1.246**|**0.797**|**0.795**|
> |Protenix ipTM|5.732|0.325|0.318|
> |NERE|3599.30|0.588|0.577|
>
> Table C. Protein-Protein Binding Affinity
> |#Method|Pearson|Spearman|RMSE|MAE|
> |:----:|:----:|:----:|:----:|:---:|
> |**ADiT-L**|**0.691**|**0.560**|**1.540**|**1.088**|
> |Protenix ipTM|0.127|0.250|1.622|2.423|
> |Rosetta|0.311|0.346|1.617|1.131|
> |DSMBind (NERE)|-|0.421|-|-|
>
> The results confirm that zero-shot baselines, while useful for structural analysis, are still far from the state-of-the-art achieved by fine-tuning models like ADiT-L.
>
> >**W3: On one hand, the author doesn't explain why they need to construct a different metric based on AlphaFold3 instead of directly using ipTM.**
>
> Thank you for raising this important point. There is a critical distinction between our task and the one addressed by ipTM. ipTM is a metric that predicts the geometric accuracy of a predicted protein complex structure. It measures how well the model thinks it built the structure, not the experimental binding energy. ADiT is designed to predict experimental binding affinity (a thermodynamic quantity like $\Delta G$ or $K_d$) from a given biomolecular structure.
>
> As we state in the introduction, "structure prediction remains an intermediate step". A high-confidence structure (high ipTM), though commonly used for binder design, does not always equate to high binding affinity, and ipTM is not designed or trained to predict this value. We have revised our related work section to explicitly discuss these zero-shot methods.

---

> > ### Author Response · Authors · 2025-11-20
> >
> > >**W4: the paper should clearly explain their difference from AF3, like what modules are reused and what modules are newly proposed in this paper.**
> >
> > We apologize if this was not sufficiently clear. We will revise Section 3.4 to make the distinction sharper. Here is a summary of the key differences:
> >
> > 1. **Goal**: AF3 is a generative model for structure prediction. ADiT is a representation learner that encodes observed geometry.
> > 2. **Inputs**: We remove all dependencies on MSAs and templates, which are critical inputs for AF3. We instead use a pre-trained language model (ESM-2) to embed sequence information.
> > 3. **Architecture**: We significantly simplify the architecture. We remove AF3's "heavy conditioning (trunk) module" and its "computationally intensive Pairformer blocks", replacing them with a more lightweight featurization scheme (detailed in Fig. 2a).
> > 4. **Re-use**: We adapt the core diffusion transformer architecture (the atom/token attention modules) , as it is "well-suited for scalability". However, we use it for representation learning via a denoising objective, not generative prediction. During fine-tuning, we remove the diffusion time-step conditioning, setting it to zero to process clean samples.
> >
> >
> > >**W5: The performance on some tasks are worse than the baseline methods like protein-protein interaction.**
> >
> > We appreciate this detailed observation. You are correct that on the protein-protein binding task (Table 2), the specialized baseline Prompt-DDG achieves marginally better results on Spearman, RMSE, and MAE metrics. However, we would like to highlight two crucial points:
> >
> > 1. On this same task, our ADiT-L model achieves the highest Pearson correlation (0.691 vs. 0.677 for Prompt-DDG), which is a key metric for this task.
> > 2. Most importantly, Prompt-DDG is a specialized model built solely for $\Delta\Delta G$ prediction in protein-protein affinity, encoding the mutation-site microenvironment and feeding this local context into downstream predictors. Its design is tightly bound to mutation effect prediction and does not extend to other interaction types such as protein-ligand or drug-target binding, where such mutation-centric contexts are absent. In contrast, ADiT is a general-purpose foundation model that learns shared representations across diverse biomolecular interaction modalities.
> >
> > The fact that our general model is highly competitive with (and even state-of-the-art on one metric) a specialized SOTA model, while also achieving state-of-the-art results on protein-ligand and drug-target binding, is a central strength of our paper. This directly supports our main contribution of creating a "generalizable framework for biomolecular interactions".
> >
> >
> > >**Q1: Can the authors explain how they curated each kind of complex data from the PDB?**
> >
> > We appreciate the reviewer's observation. In our work, interaction examples are defined at the interface level rather than at the complex level. Specifically, an "interaction example" refers to the interaction between two specific chains that form chain-chain interfaces within a single PDB entry. Two chains are considered to interact if the minimum heavy-atom distance between them is less than 5 angstroms. We will add a clarifying sentence in the manuscript to make this distinction explicit.

---

### Author Response · Authors · 2025-12-02

We thank all reviewers again for their time and thoughtful feedback. During rebuttal, we have followed the reviewers' suggestions to add more comparisons with AlphaFold3-related baselines, discussion on equivariant and invariant architectures and efficiency and interpretability analysis. We hope these responses can address the reviewres' conerns.

---

### Meta-Review · Area_Chair_vRMr · 2026-01-05

**Summary:**

This paper presents an all-atom foundation model (ADiT) for biomolecular binding affinity prediction across multiple interaction types, using a hierarchical atom and token architecture and denoising-style pretraining on noisy coordinates.
Overall, reviewers view the direction as timely and high-impact; in particular, the broad multi-task evaluation and antibody-focused analyses are compelling, and the authors’ clarifications strengthen the submission.
I recommend acceptance, with a camera-ready expectation to (i) clarify Token2Atom in enough detail to reproduce, (ii) temper “first or foundation” positioning where appropriate, and (iii) address the noted dual-use risk with a more explicit mitigation statement.

**Reviewer Concerns:**

Token2Atom implementation details remain insufficiently concrete for some reviewers.
The case for a non-equivariant design would benefit from tighter ablations (e.g., role of rotation augmentation and comparison to equivariant alternatives).
Dual-use considerations should be more explicitly discussed, per reviewer request.

**Reviewer Scores:**

nSSR (initial 8): stays 8 (explicitly indicated no score change).
kRrt (initial 8): stays ~8 (strongly positive; main remaining issue is ethics mitigation).
r7bx (initial 6): stays ~6–7 (concerns about novelty framing of evidence are partly mitigated by clarifications, but not fully eliminated).
Gvv4 (initial 4): likely increases slightly if the clarified efficiency/context and additional details are reflected clearly in the revision, but may remain cautious due to Token2Atom and non-equivariance ablation requests.

---

### Decision · Program_Chairs · 2026-01-26

Accept (Poster)